# Hodge-Aware Learning on Simplicial Complexes

## Abstract

Neural networks on simplicial complexes (SCs) can learn from data residing on simplices such as nodes, edges, triangles, etc. However, existing works often overlook the Hodge theory that decomposes simplicial data into three orthogonal characteristic subspaces, such as the identifiable gradient, curl and harmonic components of edge flows. In this paper, we aim to incorporate this data inductive bias into learning on SCs. Particularly, we present a general convolutional architecture which respects the three key principles of uncoupling the lower and upper simplicial adjacencies, accounting for the inter-simplicial couplings, and performing higher-order convolutions. To understand these principles, we first use Dirichlet energy minimizations on SCs to interpret their effects on mitigating the simplicial oversmoothing. Then, through the lens of spectral simplicial theory, we show the three principles promote the Hodge-aware learning of this architecture, in the sense that the three Hodge subspaces are invariant under its learnable functions and the learning in two nontrivial subspaces are independent and expressive. To further investigate the learning ability of this architecture, we also study it is stable against small perturbations on simplicial connections. Finally, we experimentally validate the three principles by comparing with methods that either violate or do not respect them. Overall, this paper bridges learning on SCs with the Hodge decomposition, highlighting its importance for rational and effective learning from simplicial data.

## 1   Introduction

It is not uncommon to have polyadic interactions in such as friendship networks [1], collaboration networks [2], gene regulatory networks [3], etc [4-6]. To remedy the pitfall that graphs are limited to model pairwise interactions between data entites on nodes, simplicial complexes (SCs) have become popular among others [7]. A SC can be informally viewed as an extension of a graph, which is the simplest SC, by including, not limited to, some triangles over the edge set. SCs like graphs have algebraic representations – the Hodge Laplacians, an extension of graph Laplacians [8, 9]. Moreover, besides node-wise data, SCs can support data on general simplices, e,g., flow-type data, e.g., water flows [10], traffic flows [11], information flows [12], etc., naturally arise as data on edges, and data related to three parties, e.g., triadic collaborations [2], can be defined on triangles in a SC.

Thus, existing works have built NNs on SCs to learn from such simplicial data, to name a few, [13–19]. In analogous to graph neural networks (GNNs) learning from node data relying on the adjacency between nodes, the idea behind these works is to rely on the relations between simplices. Such relations can be twofold: first, two simplices can be lower and upper adjacent to each other, such as an edge can be (lower) adjacent to another edge via a common node, and can also be (upper) adjacent to others by sharing a common triangle; and second, there exist the inter-simplicial couplings (or simplicial incidences) such that a node can induce data on its incident edge and a triangle can cause data on its three edges, or the other way around. Along with this idea, [15, 16, 19] proposed convolutional-type NNs by applying the simplicial adjacencies, [14, 20] included also inter-simplicial couplings, and [17, 21] generalized the graph message-passing [22] to SCs based on both relations.

However, these works often solely focus on the simplicial structures, overlooking *the Hodge decomposition* [23], which gives three orthogonal subspaces that uniquely characterize the simplicial data. An edge flow can be decomposed into three distinct parts: a curl-free part induced by some node data, a divergence-free (div-free) part that follows flow conservation (in-flows equal to out-flows at nodes), and a harmonic part that is both div- and curl-free. Meanwhile, real-world simplicial data often presents properties to be in certain subspaces but not others, or its components carry physical usefulness, e.g., statistical ranking, exchange market [24], traffic networks [11], brain networks [12], game theory [25], etc. Thus, intuitively, as an example, a Hodge-biased learner should not, at least not primarily, learn in the div-free space if the edge flow is curl-free. If the learning function preserves the subspaces and operates independently in three subspaces, the learning space is substantially shrunk. This in fact provides an important inductive bias allowing for rational and effective learning on SCs.

Motivated by this, in this paper, we present the general convolutional learning on SCs, SCCNN, which respects three key principles of uncoupling the lower and upper simplicial adjacencies, accounting for the inter-simplicial couplings, and performing higher-order convolutions. Unlike existing convolutional methods [14–16], which either lack theoretical insights or only discuss their architectural differences in the simplicial domain, we focus on providing a theoretical analysis of these three principles from both the perspectives of simplicial and simplicial data, specifically Hodge theory. This offers deeper and unique insights when compared to the more closely related works [19, 17].

**Main contributions.** In Section 3.2, we first use Dirichlet energy minimizations on SCs to understand how uncoupling the lower and upper adjacencies in Hodge Laplacians and the inter-simplicial couplings can mitigate the oversmoothing inherited from generalizing GCN to SCs. Under the help of spectral simplicial theory [26–28], in Section 4, we characterize the spectral behavior of SCCNN and its expressive power. We show SCCNN performs independent and expressive learning in the three subspaces of the Hodge decomposition, which are invariant under its learning operators. This Hodge-awareness (or Hodge-aided bias) allows for effective and rational learning on SCs compared to MLP or simplicial message-passing [17]. In Section 5, we also prove it is stable against small perturbations on the strengths of simplicial connections, and show how three principles can affect the stability. Lastly, we validate our findings on different simplicial tasks, including recovering foreign currency exchange (forex) rates, predicting triadic and tetradic collaborations, and trajectories.

## 2    Background

**Simplicial complex and simplicial adjacencies.** A $k$-simplex $s^k$ is a subset of $\mathcal{V} = \{1, \ldots, n_0\}$ with cardinality $k + 1$. A *face* of $s^k$ is a subset with cardinality $k$. A *coface* of $s^k$ is a $(k + 1)$-simplex that has $s^k$ as a face. Nodes, edges and (filled) triangles are geometric realizations of 0-, 1- and 2-simplices. A SC $\mathcal{S}$ of order $K$ is a collection of $k$-simplices, $k = 0, \ldots, K$, with the *inclusion* property: $s^{k-1} \in \mathcal{S}$ if $s^{k-1} \subset s^k$ for $s^k \in \mathcal{S}$. A graph is a SC of order one and by taking into account some triangles, we obtain a SC of order two. We collect all $k$-simplices of $\mathcal{S}$ in set $\mathcal{S}^k = \{s_i^k\}_{i=1,\ldots,n_k}$ with $n_k = |\mathcal{S}^k|$, i.e., $\mathcal{S} = \cup_{k=0}^{K} \mathcal{S}^k$. For $s^k$, We say a $k$-simplex is *lower (upper) adjacent* to $s^k$ if they share a common face (coface). For computations, an *orientation* of a simplex is chosen as an ordering of its vertices (a node has a trivial orientation). Here we consider the lexicographical ordering $s^k = [1, \ldots, k + 1]$, e.g., a triangle $s^2 = \{i, j, k\}$ is oriented as $[i, j, k]$.

**Algebraic representation.** Incidence matrix $\mathbf{B}_k$ describes the relations between $(k - 1)$- (i.e., faces) and $k$-simplices, e.g., $\mathbf{B}_1$ is the node-to-edge incidence matrix and $\mathbf{B}_2$ edge-to-triangle. We have $\mathbf{B}_k \mathbf{B}_{k+1} = \mathbf{0}$ by definition [9]. The $k$-*Hodge Laplacian* is $\mathbf{L}_k = \mathbf{B}_k^\top \mathbf{B}_k + \mathbf{B}_{k+1} \mathbf{B}_{k+1}^\top$ with the *lower Laplacian* $\mathbf{L}_{k,\mathrm{d}} = \mathbf{B}_k^\top \mathbf{B}_k$ and the *upper Laplacian* $\mathbf{L}_{k,\mathrm{u}} = \mathbf{B}_{k+1} \mathbf{B}_{k+1}^\top$. We have a set of $\mathbf{L}_k, k = 1, \ldots, K - 1$ in a SC of order $K$ with the graph Laplacian $\mathbf{L}_0 = \mathbf{B}_1 \mathbf{B}_1^\top$, and $\mathbf{L}_K = \mathbf{B}_K^\top \mathbf{B}_K$. Note that $\mathbf{L}_{k,\mathrm{d}}$ and $\mathbf{L}_{k,\mathrm{u}}$ encode the lower and upper adjacencies of $k$-simplices. For example, for $k = 1$, they encode the edge-to-edge adjacencies through nodes and triangles, respectively.

**Simplicial data.** A $k$-*simplicial data (or $k$-signal)* $\mathbf{x}_k \in \mathbb{R}^{n_k}$ is defined by an *alternating* map $f_k$ which assigns a real value to a simplex, and restricts that if the orientation of a simplex is anti-aligned with the reference orientation, then the sign of the signal value will be changed [9].

**Incidence matrices as derivative operators on SCs.** We can measure how a $k$-signal $\mathbf{x}_k$ varies w.r.t. the faces and cofaces of $k$-simplices by applying $\mathbf{B}_k \mathbf{x}_k$ and $\mathbf{B}_{k+1}^\top \mathbf{x}_k$ [29]. For a node signal $\mathbf{x}_0$, $\mathbf{B}_1^\top \mathbf{x}_0$ computes its *gradient* as the difference between adjacent nodes. Thus, a constant $\mathbf{x}_0$ has zero gradient. For an edge flow $\mathbf{x}_1$, $[\mathbf{B}_1 \mathbf{x}_1]_j = \sum_{i<j} [\mathbf{x}_1]_{[i,j]} - \sum_{j<k} [\mathbf{x}_1]_{[j,k]}$ computes its *divergence*, which is the

94 difference between the in-flow and the out-flow at node $j$, and $[\mathbf{B}_2^\top \mathbf{x}_1]_t = [\mathbf{x}_1]_{[i,j]} - [\mathbf{x}_1]_{[i,k]} + [\mathbf{x}_1]_{[j,k]}$

95 computes the *curl* of $\mathbf{x}_1$, which is the net-flow circulation in triangle $t = [i, j, k]$.

96 **Theorem 1** (Hodge decomposition [23, 9])**.** *The $k$-simplicial data space admits an orthogonal direct*

97 *sum decomposition $\mathbb{R}^{n_k} = \text{im}(\mathbf{B}_k^\top) \oplus \ker(\mathbf{L}_k) \oplus \text{im}(\mathbf{B}_{k+1})$. Moreover, we have $\ker(\mathbf{B}_{k+1}^\top) =$*

98 $\text{im}(\mathbf{B}_k^\top) \oplus \ker(\mathbf{L}_k)$ *and* $\ker(\mathbf{B}_k) = \ker(\mathbf{L}_k) \oplus \text{im}(\mathbf{B}_{k+1})$.

99 In the node space, this is trivial as $\mathbb{R}^{n_0} = \ker(\mathbf{L}_0) \oplus \text{im}(\mathbf{B}_1)$ where the kernel of $\mathbf{L}_0$ contains constant

100 node data and the image of $\mathbf{B}_1$ contains nonconstant data. In the edge case, three subspaces carry

101 more tangible meaning: the *gradient space* $\text{im}(\mathbf{B}_1^\top)$ collects edge flows as the gradient of some node

102 signal, which are *curl-free*; the *curl* space $\text{im}(\mathbf{B}_2)$ consists of flows cycling around triangles, which

103 are *div-free*; and flows in the *harmonic space* $\ker(\mathbf{L}_1)$ are both div- and curl-free. In this paper, we

104 inherit the names of three edge subspaces to general $k$-simplices. This theorem states that any $\mathbf{x}_k$ can

105 be uniquely expressed as $\mathbf{x}_k = \mathbf{x}_{k,\text{G}} + \mathbf{x}_{k,\text{H}} + \mathbf{x}_{k,\text{C}}$ with gradient part $\mathbf{x}_{k,\text{G}} = \mathbf{B}_k^\top \mathbf{x}_{k-1}$, curl part

106 $\mathbf{x}_{k,\text{C}} = \mathbf{B}_{k+1} \mathbf{x}_{k+1}$, for some $\mathbf{x}_{k\pm1}$, and harmonic part following $\mathbf{L}_k \mathbf{x}_{k,\text{H}} = \mathbf{0}$.

## 3 Simplicial Complex CNNs

108 We start by introducing the general convolutional architecture on SCs, followed by its properties, then

109 we discuss its components from an energy minimizations perspective. We refer to Appendix B for

110 some illustrations. In a SC, a SCCNN at layer $l$ computes the $k$-output $\mathbf{x}_k^l$ with $\mathbf{x}_{k-1}^{l-1}, \mathbf{x}_k^{l-1}$ and $\mathbf{x}_{k+1}^{l-1}$

111 as inputs, i.e., a map $\text{SCCNN}_k^l : \{\mathbf{x}_{k-1}^{l-1}, \mathbf{x}_k^{l-1}, \mathbf{x}_{k+1}^{l-1}\} \to \mathbf{x}_k^l$, for all $k$. It admits a detailed form

$$\mathbf{x}_k^l = \sigma(\mathbf{H}_{k,\text{d}}^l \mathbf{x}_{k,\text{d}}^{l-1} + \mathbf{H}_k^l \mathbf{x}_k^{l-1} + \mathbf{H}_{k,\text{u}}^l \mathbf{x}_{k,\text{u}}^{l-1}), \text{ with } \mathbf{H}_k^l = \sum_{t=0}^{T_\text{d}} w_{k,\text{d},t}^l \mathbf{L}_{k,\text{d}}^t + \sum_{t=0}^{T_\text{u}} w_{k,\text{u},t}^l \mathbf{L}_{k,\text{u}}^t. \quad (1)$$

112 1) Previous output $\mathbf{x}_k^{l-1}$ is passed to a simplicial convolution filter (SCF [30]) $\mathbf{H}_k^l$ of orders $T_\text{d}, T_\text{u}$,

113 which performs a linear combination of the data from up to $T_\text{d}$-hop lower-adjacent and $T_\text{u}$-hop

114 upper-adjacent simplices, weighted by two sets of learnable weights $\{w_{k,\text{d},t}^l\}, \{w_{k,\text{u},t}^l\}$.

115 2) $\mathbf{x}_{k,\text{d}}^{l-1} = \mathbf{B}_k^\top \mathbf{x}_{k-1}^{l-1}$ and $\mathbf{x}_{k,\text{u}}^{l-1} = \mathbf{B}_{k+1} \mathbf{x}_{k+1}^{l-1}$ are the lower and upper projections from $(k \pm 1)$-

116 simplices via incidence relations, respectively. Then, $\mathbf{x}_{k,\text{d}}^{l-1}$ is passed to a lower SCF $\mathbf{H}_{k,\text{d}}^l :=$

117 $\sum_{t=0}^{T_\text{d}} w_{k,\text{d},t}'^l \mathbf{L}_{k,\text{d}}^t$, and the upper projection $\mathbf{x}_{k,\text{u}}^{l-1}$ is passed to an upper SCF $\mathbf{H}_{k,\text{u}}^l := \sum_{t=0}^{T_\text{u}} w_{k,\text{u},t}'^l \mathbf{L}_{k,\text{u}}^t$.

118 Lastly, the sum of the three SCF outputs is passed to an elementwise nonlinearity $\sigma(\cdot)$.

119 This architecture subsumes the methods in [14–16, 19, 18, 20]. Particularly, we emphasize on the key

120 three principles. 1) Uncouple the lower and upper Laplacians: this leads to an independent treatment

121 of the lower and upper adjacencies, achieved by using two sets of learnable weights; 2) Account for

122 the inter-simplicial couplings: $\mathbf{x}_{k,\text{d}}$ and $\mathbf{x}_{k,\text{u}}$ carry the nontrivial information contained in the faces

123 and cofaces (by Theorem 1); and 3) Perform higher-order convolutions: considering $T_\text{d}, T_\text{u} \geq 1$ in

124 SCFs which leads to a multi-hop receptive field on SCs. In short, SCCNN propagates information

125 across SCs based on two simplicial adjacencies and two incidences in a multi-hop fashion.

### 3.1 Properties

127 **Simplicial locality.** SCFs admit an intra-simplicial locality: $\mathbf{H}_k \mathbf{x}_k$ is localized in $T_\text{d}$-hop lower and

128 $T_\text{u}$-hop upper $k$-simplicial neighborhoods [30]. A SCCNN preserves such locality as $\sigma(\cdot)$ does not

129 alter the information locality. It also admits the inter-simplicial locality between $k$- and $(k \pm 1)$-

130 simplices, which extends to simplices of orders $k \pm 2$ if $L \geq 2$ because $\mathbf{B}_k \sigma(\mathbf{B}_{k+1}) \neq \mathbf{0}$ [31].

131 Moreover, the two localities are coupled in a multi-hop way through SCFs such that a node not only

132 interacts with its incident edges and the triangles including it, but also those further hops away.

133 **Complexity.** A SCCNN layer has the parameter complexity of order $\mathcal{O}(T_\text{d} + T_\text{u})$ and the computa-

134 tional complexity $\mathcal{O}(k(n_k + n_{k+1}) + n_k m_k (T_\text{d} + T_\text{u}))$, linear to the simplex dimensions, where $m_k$

135 is the maximum of the number of neighbors for $k$-simplices.

136 **Equivariance.** SCCNNs are permutation-equivariant, which allows us to list simplices in any order,

137 and orinetation-equivariant if $\sigma(\cdot)$ is odd, which gives us the freedom to choose reference orientations.

138 In Appendix B.3, we provide formal discussions on such equivariances and why permutations form a

139 symmetry group of a SC and orientation changes are symmetries of data space but not SCs.

### 3.2 A perspective of SCCNN from Dirichlet energy minimization on SCs

141 **Definition 2.** The *Dirichlet energy* of $\mathbf{x}_k$ is $D(\mathbf{x}_k) = D_\text{d}(\mathbf{x}_k) + D_\text{u}(\mathbf{x}_k) := \|\mathbf{B}_k \mathbf{x}_k\|_2^2 + \|\mathbf{B}_{k+1}^\top \mathbf{x}_k\|_2^2$.

142 For node signals, $D(\mathbf{x}_0) = \|\mathbf{B}_1^\top \mathbf{x}_0\|_2^2 = \sum_i \sum_j \|x_{0,i} - x_{0,j}\|^2$ is a $\ell_2$-norm of the *gradient* of $\mathbf{x}_0$.
143 For edge flows, $D(\mathbf{x}_1)$ is the sum of the total divergence and curl, which measure the flow variations
144 w.r.t. nodes and triangles, respectively. In general, $D(\mathbf{x}_k)$ measures the lower and upper $k$-simplicial
145 signal variations w.r.t. the faces ($D_{\mathrm{d}}(\mathbf{x}_k)$) and cofaces ($D_{\mathrm{u}}(\mathbf{x}_k)$). A $k$-signal $\mathbf{x}_k$ with $D(\mathbf{x}_k) = 0$)
146 follows $\mathbf{L}_k \mathbf{x}_k = \mathbf{0}$, called *harmonic*, e.g., a constant node signal and a div- and curl-free edge flow.

147 **Simplicial shifting as Hodge Laplacian smoothing.** [14, 20] considered $\mathbf{H}_k$ as a weighted variant
148 of $\mathbf{I} - \mathbf{L}_k$, generalizing the GCN layer [32]. This simplicial shifting step is necessarily a Hodge
149 Laplacian smoothing [31]. Given an initial $\mathbf{x}_k^0$, we consider the Dirichlet energy minimization:

$$\min_{\mathbf{x}_k} \|\mathbf{B}_k \mathbf{x}_k\|_2^2 + \gamma\|\mathbf{B}_{k+1}^\top \mathbf{x}_k\|_2^2, \gamma > 0, \text{ gradient descent: } \mathbf{x}_{k,\mathrm{gd}}^{l+1} = (\mathbf{I} - \eta\mathbf{L}_{k,\mathrm{d}} - \eta\gamma\mathbf{L}_{k,\mathrm{u}})\mathbf{x}_k^l \quad (2)$$

150 with step size $\eta > 0$. The simplicial shifting $\mathbf{x}_k^{l+1} = w_0(\mathbf{I} - \mathbf{L}_k)\mathbf{x}_k^l$ is a gradient descent with
151 $\eta = \gamma = 1$ and weighted by $w_0$, then followed by nonlinearity. A minimizer of Eq. (2) with $\gamma = 1$ is
152 in the harmonic space $\ker(\mathbf{L}_k)$. Thus, an NN composed of simplicial shifting layers may lead to an
153 output with exponentially decreasing Dirichlet energy as it deepens, i.e., *simplicial oversmoothing*.

154 **Proposition 3.** *If $w_0^2\|\mathbf{I} - \mathbf{L}_k\|_2^2 < 1$, $D(\mathbf{x}_k^{l+1})$ in a simplicial shifting exponentially converges to 0.*

155 This generalizes the oversmoothing of GCN and its variants [33–35]. However, when uncoupling
156 the lower and upper parts of $\mathbf{L}_k$ in this shifting, associated with $\gamma \neq 1$, the decrease of $D(\mathbf{x}_k)$ can
157 slow down or cease because the objective instead looks for a solution primarily in either $\ker(\mathbf{B}_k)$
158 (for $\gamma \ll 1$) or $\ker(\mathbf{B}_{k+1}^\top)$ (for $\gamma \gg 1$), not necessarily in $\ker(\mathbf{L}_k)$, as we show in Section 6.

159 **Inter-simplicial couplings as sources.** Given some nontrivial $\mathbf{x}_{k-1}$ and $\mathbf{x}_{k+1}$, we consider

$$\min_{\mathbf{x}_k} \|\mathbf{B}_k \mathbf{x}_k - \mathbf{x}_{k-1}\|_2^2 + \|\mathbf{B}_{k+1}^\top \mathbf{x}_k - \mathbf{x}_{k+1}\|_2^2, \quad (3)$$

160 which has a gradient descent $\mathbf{x}_{k,\mathrm{gd}}^{l+1} = (\mathbf{I} - \eta\mathbf{L}_k)\mathbf{x}_k^l + \eta(\mathbf{x}_{k,\mathrm{d}} + \mathbf{x}_{k,\mathrm{u}})$. It resembles the whole layer
161 in [14, 20], $\mathbf{x}_k^{l+1} = w_0(\mathbf{I} - \mathbf{L}_k)\mathbf{x}_k^l + w_1\mathbf{x}_{k,\mathrm{d}} + w_2\mathbf{x}_{k,\mathrm{u}}$ with some weights, followed by nonlinearity.
162 We have $D(\mathbf{x}_k^{l+1}) \leq w_0^2\|\mathbf{I} - \mathbf{L}_k\|_2^2 D(\mathbf{x}_k^l) + w_1^2\lambda_{\max}(\mathbf{L}_{k,\mathrm{d}})\|\mathbf{x}_{k,\mathrm{d}}\|_2^2 + w_2^2\lambda_{\max}(\mathbf{L}_{k,\mathrm{u}})\|\mathbf{x}_{k,\mathrm{u}}\|_2^2$, by
163 triangle inequality. The projections here act as energy sources, and also the objective looks for an
164 $\mathbf{x}_k$ in the images of $\mathbf{B}_{k+1}$ and $\mathbf{B}_k^\top$, instead of $\ker(\mathbf{L}_k)$ when $\mathbf{x}_{k-1}$ and $\mathbf{x}_{k+1}$ are not trivial. Thus,
165 inter-simplicial couplings can potentially mitigate the oversmoothing as well.

166 Here we show simply generalzing GCN will inherit its oversmoothing to SCs. However, both the
167 separation of the lower and upper Laplacians and inter-simplicial couplings could potentially mitigate
168 this oversmoothing. We here considered a Dirichlet energy minimization perspective. They can also
169 be explained by means of diffusion process on SCs [36]. We refer to Appendix B.4 for this.

## 170 4 From convolutional to Hodge-aware

171 In this section, we show how SCCNN, guided by the three principles, performs *the Hodge-aware*
172 *learning*, allowing for rational and effective learning on SCs while remaining expressive. To ease the
173 exposition, we first provide a more fine-grained spectral view on how SCCNN learns from simplicial
174 data of different variations in the three subspaces based on the simplicial spectral theory [27, 26, 30].
175 Then, we characterize its expressive power and discuss its Hodge-awareness.

176 **Definition 4** ([27]). *The simplicial Fourier transform (SFT) of $\mathbf{x}_k$ is $\tilde{\mathbf{x}}_k = \mathbf{U}_k^\top \mathbf{x}_k$ where the Fourier*
177 *basis $\mathbf{U}_k$ can be found as the eigenbasis of $\mathbf{L}_k$ and the eigenvalues are simplicial frequencies.*

178 **Proposition 5** ([26]). *The SFT basis can be found as $\mathbf{U}_k = [\mathbf{U}_{k,\mathrm{H}} \ \mathbf{U}_{k,\mathrm{G}} \ \mathbf{U}_{k,\mathrm{C}}]$ where 1) the zero*
179 *eigenspace $\mathbf{U}_{k,\mathrm{H}}$ of $\mathbf{L}_k$ spans $\ker(\mathbf{L}_k)$, and an eigenvalue $\lambda_{k,\mathrm{H}} = 0$ is a harmonic frequency; 2) the*
180 *nonzero eigenspace $\mathbf{U}_{k,\mathrm{G}}$ of $\mathbf{L}_{k,\mathrm{d}}$ spans $\mathrm{im}(\mathbf{B}_k^\top)$, and an eigenvalue $\lambda_{k,\mathrm{G}}$ is a gradient frequency,*
181 *measuring the lower variation $D_{\mathrm{d}}(\mathbf{u}_{k,\mathrm{G}})$; 3) the nonzero eigenspace $\mathbf{U}_{k,\mathrm{C}}$ of $\mathbf{L}_{k,\mathrm{u}}$ spans $\mathrm{im}(\mathbf{B}_{k+1})$,*
182 *and an eigenvalue $\lambda_{k,\mathrm{C}}$ is a curl frequency, measuring the upper variation $D_{\mathrm{u}}(\mathbf{u}_{k,\mathrm{C}})$.*

183 Thus, the SFT of $\mathbf{x}_k$ can be found as $\tilde{\mathbf{x}}_k = [\tilde{\mathbf{x}}_{k,\mathrm{H}}^\top, \tilde{\mathbf{x}}_{k,\mathrm{G}}^\top, \tilde{\mathbf{x}}_{k,\mathrm{C}}^\top]^\top$, where each component is the
184 intensity of $\mathbf{x}_k$ at a simplicial frequency. Consider $\mathbf{y}_k = \mathbf{H}_{k,\mathrm{d}}\mathbf{x}_{k,\mathrm{d}} + \mathbf{H}_k\mathbf{x}_k + \mathbf{H}_{k,\mathrm{u}}\mathbf{x}_{k,\mathrm{u}}$ in a SCCNN
185 layer. Multiplying on both sides by $\mathbf{U}_k$, we then have the SFT $\tilde{\mathbf{y}}$ as

$$\begin{cases} \tilde{\mathbf{y}}_{k,\mathrm{H}} = \tilde{\mathbf{h}}_{k,\mathrm{H}} \odot \tilde{\mathbf{x}}_{k,\mathrm{H}}, \\ \tilde{\mathbf{y}}_{k,\mathrm{G}} = \tilde{\mathbf{h}}_{k,\mathrm{d}} \odot \tilde{\mathbf{x}}_{k,\mathrm{d}} + \tilde{\mathbf{h}}_{k,\mathrm{G}} \odot \tilde{\mathbf{x}}_{k,\mathrm{G}}, \\ \tilde{\mathbf{y}}_{k,\mathrm{C}} = \tilde{\mathbf{h}}_{k,\mathrm{C}} \odot \tilde{\mathbf{x}}_{k,\mathrm{C}} + \tilde{\mathbf{h}}_{k,\mathrm{u}} \odot \tilde{\mathbf{x}}_{k,\mathrm{u}}, \end{cases} \text{where} \begin{cases} \tilde{\mathbf{h}}_{k,\mathrm{H}} = (w_{k,\mathrm{d},0} + w_{k,\mathrm{u},0})\mathbf{1}, \\ \tilde{\mathbf{h}}_{k,\mathrm{G}} = \sum_{t=0}^{T_{\mathrm{d}}} w_{k,\mathrm{d},t}\boldsymbol{\lambda}_{k,\mathrm{G}}^{\odot t} + w_{k,\mathrm{u},0}\mathbf{1}, \\ \tilde{\mathbf{h}}_{k,\mathrm{C}} = \sum_{t=0}^{T_{\mathrm{u}}} w_{k,\mathrm{u},t}\boldsymbol{\lambda}_{k,\mathrm{C}}^{\odot t} + w_{k,\mathrm{d},0}\mathbf{1}, \end{cases} \quad (4)$$

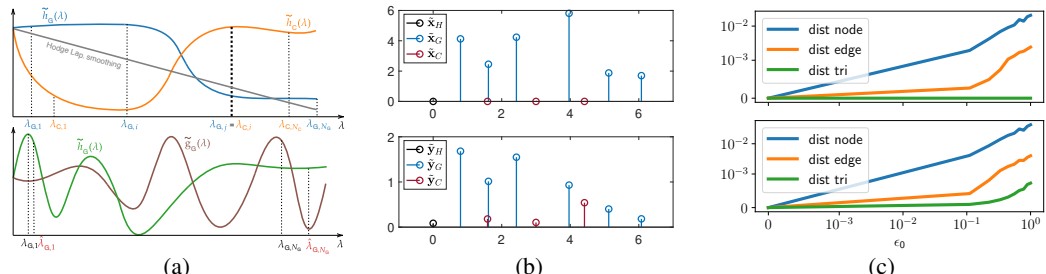

Figure 1: (a) *(top)*: Independent gradient and curl learning responses. *(bottom)*: Stability-selectivity tradeoff of SCFs where $\tilde{h}_G$ has better stability but smaller selectivity than $\tilde{g}_G$. (b) Information spillage of nonlinearity. (c) The distance between the perturbed outputs and true when node adjacencies are perturbed. *(top)*: $L = 1$, triangle output remains clean. *(bottom)*: $L = 2$, triangle output is perturbed.

is the frequency response of $\mathbf{H}_k$ as $\tilde{\mathbf{h}}_k = \mathrm{diag}(\mathbf{U}_k^\top \mathbf{H}_k \mathbf{U}_k)$ [30], and $\tilde{\mathbf{h}}_{k,\mathrm{d}}$ and $\tilde{\mathbf{h}}_{k,\mathrm{u}}$, the responses of $\mathbf{H}_{k,\mathrm{d}}$ and $\mathbf{H}_{k,\mathrm{u}}$, can be expressed accordingly. This spectral relation (4) shows how the learning of SCCNN is performed at frequencies in different subspaces. Specifically, the gradient SFT $\tilde{\mathbf{x}}_{k,\mathrm{G}}$ is learned by a gradient response $\tilde{\mathbf{h}}_{k,\mathrm{G}}$, which is independent of the curl response $\tilde{\mathbf{h}}_{k,\mathrm{C}}$ learning the curl SFT $\tilde{\mathbf{x}}_{k,\mathrm{C}}$, and they only coincide at the trivial harmonic frequency, as shown in Fig. 1a. Likewise, the lower and upper projections are independently learned by $\tilde{\mathbf{h}}_{k,\mathrm{d}}$ and $\tilde{\mathbf{h}}_{k,\mathrm{u}}$, respectively.

The nonlinearity induces the information spillage that one type of spectra could be spread over other types. That is, $\sigma(\tilde{\mathbf{y}}_{k,\mathrm{G}})$ could contain information in harmonic or curl subspaces, as illustrated in Fig. 1b. This is to increase the expressive power of SCCNN, which can be characterized as follows.

**Theorem 6.** *A SCCNN layer with inputs $\mathbf{x}_{k,\mathrm{d}}, \mathbf{x}_k, \mathbf{x}_{k,\mathrm{u}}$ is at most expressive as an MLP layer $\sigma(\mathbf{G}'_{k,\mathrm{d}}\mathbf{x}_{k,\mathrm{d}} + \mathbf{G}_k\mathbf{x}_k + \mathbf{G}'_{k,\mathrm{u}}\mathbf{x}_{k,\mathrm{u}})$ with $\mathbf{G}_k = \mathbf{G}_{k,\mathrm{d}} + \mathbf{G}_{k,\mathrm{u}}$ where $\mathbf{G}_{k,\mathrm{d}}$ and $\mathbf{G}_{k,\mathrm{u}}$ are analytical matrix functions of $\mathbf{L}_{k,\mathrm{d}}$ and $\mathbf{L}_{k,\mathrm{u}}$, respectively, and $\mathbf{G}'_{k,\mathrm{d}}$ and $\mathbf{G}'_{k,\mathrm{u}}$ likewise. Moreover, this expressivity can be achieved when setting $T_\mathrm{d} = T'_\mathrm{d} = n_{k,\mathrm{G}}$ and $T_\mathrm{u} = T'_\mathrm{u} = n_{k,\mathrm{C}}$ in Eq. (1) with $n_{k,\mathrm{G}}$ the number of distinct gradient frequencies and $n_{k,\mathrm{C}}$ the number of distinct curl frequencies.*

The proof follows from Cayley-Hamilton theorem [37]. This expressive power can be better understood spectrally. The gradient SFT of $\mathbf{x}_k$ can be learned most expressively by an analytical function $g_{k,\mathrm{G}}(\lambda)$, the eigenvalue of $\mathbf{G}_{k,\mathrm{d}}$ at a gradient frequency. And the curl SFT of $\mathbf{x}_k$ can be learned most expressively by another analytical function $g_{k,\mathrm{C}}(\lambda)$, the eigenvalue of $\mathbf{G}_{k,\mathrm{u}}$ at a curl frequency. These two functions only need to coincide at harmonic frequency $\lambda = 0$. The SFTs of lower and upper projections can be learned most expressively by two independent functions as well. Given this expressive power and Eq. (4), we show SCCNN performs the Hodge-aware learning as follows.

**Theorem 7.** *A SCCNN is Hodge-aware in the sense that 1) three Hodge subspaces are **invariant** under the learnable SCF $\mathbf{H}_k$, i.e., $\mathbf{H}_k\mathbf{x} \in \mathrm{im}(\mathbf{B}_k^\top)$ if $\mathbf{x} \in \mathrm{im}(\mathbf{B}_k^\top)$, and likewise for $\mathrm{im}(\mathbf{B}_{k+1}), \ker(\mathbf{L}_k)$; 2) the gradient and curl spaces are invariant under the learnable lower SCF $\mathbf{H}_{k,\mathrm{d}}$ and upper SCF $\mathbf{H}_{k,\mathrm{u}}$, respectively; 3) the learning in the gradient and curl spaces are **independent and expressive**.*

This theorem essentially shows SCCNN performs expressive learning independently in the gradient and curl subspaces from three inputs while preserving the three subspaces to be invariant w.r.t its learning functions. This allows for the **rational and effective learning** on SCs. On one hand, the invariance of subspaces under the learnable SCFs substantially shrinks the learning space and makes SCCNN effective, meanwhile, its expressive power is guaranteed by the independent expressive learners, together with the nonlinearity. Instead, the non-Hodge-aware learning, e.g., MLP or simplicial message-passing using MLP to aggregate and update [17], has a much larger learning space which requires more training data for accurate learning, as well as larger computational complexity.

On the other hand, simplicial data often presents (implicit or explicit) properties that Hodge subspaces can capture. For example, water flows, traffic flows, electric currents [29, 11] follow flow conservation (div-free, in $\ker(\mathbf{B}_1)$), or curl-free forex rates, as we show in Section 6, or the gradient component of pairwise comparison data gives consistent global ranking but others are unwanted [24]. SCCNN is able to capture these characteristics effectively, generating rational outputs due to the invariance of subspaces and independent learning in gradient and curl spaces. We illustrate a trivial example below.

*Example* 8. Suppose learning to remove non-div-free noise from some input for flow conservation. SCCNN can correctly do so because when a not-well-learned SCF, preserving the noise and useful

parts primarily in their own spaces, causes large loss, e.g., mse, the Hodge-awareness restricts it to suppress in the gradient space and preserve in others. This however can be difficult non-Hodge-aware learners, e.g., MLP or MPSN [17], especially when the amount of data is limited, because the non-div-free noise can be disguised as useful by their unbiased transformation into other spaces, and the useful parts could be transformed into noise space, generating irrational non-div-free output though the overall mse can be small. *Thus, simplicial data characteristics can be easily ignored by non-Hodge-aware learners when the invariance condition is not satisfied.*

**Comparison to others.** We here discuss some other existing learning methods on SCs to emphasize on the Hodge-awareness. [15] considered $\mathbf{H}_k = \sum_i w_i \mathbf{L}_k^i$ to perform convolutions without uncoupling the lower and upper parts of $\mathbf{L}_k$, which makes it *strictly less expressive* and non-Hodge-aware, because it cannot perform different learning at frequencies in both gradient and curl spaces, though deeper layers and higher orders can compensate its expressive at other frequencies. [16] applied $\mathbf{H}_k$ with $T_\mathrm{d} = T_\mathrm{u} = 1$, which has a limited linear learning response. SCCNN returns the methods in [19, 38] when there is no inter-simplicial coupling needed. [14, 20] took the form of simplicial shifting by generalizing the GCN without uncoupling the two adjacencies, which is not-Hodge-aware. Spectrally, this gives a limited lower-pass linear spectral response, shown in Fig. 1a.

## 5 How robust are SCCNNs to domain perturbations?

In practice, a SCCNN is often built on a weighted SC to capture the strengths of simplicial adjacencies and incidences, with a same form as Eq. (1), except for that the Hodge Laplacians and the projection matrices are weighted, denoted as general operators $\mathbf{R}_{k,\mathrm{d}}, \mathbf{R}_{k,\mathrm{u}}$. These matrices are often defined following [29, 39, 40], e.g., [14, 20] considered a particular random walk formulation [41], or can be learned from data, e.g., via an attention method [42, 38]. Since SCCNN relies on the Hodge Laplacians and projection matrices, in this section, we address the question, *when these operators are perturbed, how accurate and robust are the outputs of a SCCNN?* This models the domain perturbations on the strengths of adjacent and incident relations such as a large weight is applied when two edges are weakly or not adjacent, or data on a node projects on an edge not incident to it. By quantifying this stability, we can explain the robust learning ability of SCCNN. We consider a relative perturbation model, also used to study the stability of CNNs [43–45] and GNNs [46–49].

Denote the perturbed lower and upper Laplacians as $\widehat{\mathbf{L}}_{k,\mathrm{d}}$ and $\widehat{\mathbf{L}}_{k,\mathrm{u}}$ by perturbations $\mathbf{E}_{k,\mathrm{d}}$ and $\mathbf{E}_{k,\mathrm{u}}$, and the lower and upper projections as $\widehat{\mathbf{R}}_{k,\mathrm{d}}$ and $\widehat{\mathbf{R}}_{k,\mathrm{u}}$ by perturbations $\mathbf{J}_{k,\mathrm{d}}$ and $\mathbf{J}_{k,\mathrm{u}}$, respectively.

**Definition 9** (Relative perturbation). Consider some perturbation matrix $\mathbf{E}$ of an appropriate dimension. For a symmetric matrix $\mathbf{A}$, its (relative) perturbed version is $\widehat{\mathbf{A}}(\mathbf{E}) = \mathbf{A} + \mathbf{E}\mathbf{A} + \mathbf{A}\mathbf{E}$. For a rectangular matrix $\mathbf{B}$, its (relative) perturbed version is $\widehat{\mathbf{B}}(\mathbf{E}) = \mathbf{B} + \mathbf{E}\mathbf{B}$.

This relative perturbation model, in contrast to an absolute one [47], quantifies perturbations w.r.t. the local simplicial topology in the sense that weaker connections in a SC are deviated by perturbations proportionally less than stronger connections. We further consider the integral Lipschitz property, extended from [47], to measure the variability of spectral response functions of $\mathbf{H}_k$.

**Definition 10.** A SCF $\mathbf{H}_k$ is *integral Lipschitz* with constants $c_{k,\mathrm{d}}, c_{k,\mathrm{u}} \geq 0$ if the derivatives of response functions $\tilde{h}_{k,\mathrm{G}}(\lambda)$ and $\tilde{h}_{k,\mathrm{C}}(\lambda)$ follow that $|\lambda \tilde{h}'_{k,\mathrm{G}}(\lambda)| \leq c_{k,\mathrm{d}}$ and $|\lambda \tilde{h}'_{k,\mathrm{C}}(\lambda)| \leq c_{k,\mathrm{u}}$.

This property provides a stability-selectivity tradeoff of SCFs independently in gradient and curl frequencies. A spectral response can have both good selectivity and stability in small frequencies (a large $|\tilde{h}'_{k,\cdot}|$ for $\lambda \to 0$), while in large frequencies, it tends to be flat for better stability at the cost of selectivity (a small variability for large $\lambda$), as shown in Fig. 1a. As of the polynomial nature of responses, all SCFs of a SCCNN are integral Lipschitz. We also denote the integral Lipschitz constant for the lower SCFs $\mathbf{H}_{k,\mathrm{d}}$ by $c_{k,\mathrm{d}}$ and for the upper SCFs $\mathbf{H}_{k,\mathrm{u}}$ by $c_{k,\mathrm{u}}$. Given the following reasonable assumptions, we are ready to characterize the stability bound of a SCCNN.

**Assumption 11.** *a) The perturbations are small such that* $\|\mathbf{E}_{k,\mathrm{d}}\|_2 \leq \epsilon_{k,\mathrm{d}}, \|\mathbf{J}_{k,\mathrm{d}}\|_2 \leq \varepsilon_{k,\mathrm{d}}, \|\mathbf{E}_{k,\mathrm{u}}\|_2 \leq \epsilon_{k,\mathrm{u}}$ *and* $\|\mathbf{J}_{k,\mathrm{u}}\|_2 \leq \varepsilon_{k,\mathrm{u}}$. *b) The SCFs* $\mathbf{H}_k$ *of a SCCNN have a normalized bounded frequency response (for simplicity, though unnecessary), likewise for* $\mathbf{H}_{k,\mathrm{d}}$ *and* $\mathbf{H}_{k,\mathrm{u}}$. *c) The lower and upper projections are finite* $\|\mathbf{R}_{k,\mathrm{d}}\|_2 \leq r_{k,\mathrm{d}}$ *and* $\|\mathbf{R}_{k,\mathrm{u}}\|_2 \leq r_{k,\mathrm{u}}$. *d) The nonlinearity* $\sigma(\cdot)$ *is* $c_\sigma$*-Lipschitz (e.g.,* relu, tanh, sigmoid*). e) The initial input* $\mathbf{x}_k^0$, *for all* $k$, *is finite,* $\|\mathbf{x}_k^0\|_2 \leq [\boldsymbol{\beta}]_k$, .

**Theorem 12.** *Let* $\mathbf{x}_k^L$ *be the* $k$*-simplicial output of an* $L$*-layer SCCNN on a weighted SC. Let* $\hat{\mathbf{x}}_k^L$ *be the output of the same SCCNN but on a relatively perturbed SC. Under Assumption 11, the Euclidean*

distance between the two outputs is finite and upper-bounded $\|\hat{\mathbf{x}}_k^L - \mathbf{x}_k^L\|_2 \le [\mathbf{d}]_k$ where

$$\mathbf{d} = c_\sigma^L \sum_{l=1}^{L} \widehat{\mathbf{Z}}^{l-1} \mathbf{T} \mathbf{Z}^{L-l} \boldsymbol{\beta}, \text{ with, e.g., } \mathbf{T} = \begin{bmatrix} t_0 & t_{0,\mathrm{u}} \\ t_{1,\mathrm{d}} & t_1 & t_{1,\mathrm{u}} \\ & t_{2,\mathrm{d}} & t_2 \end{bmatrix} \mathbf{Z} = \begin{bmatrix} 1 & r_{0,\mathrm{u}} \\ r_{1,\mathrm{d}} & 1 & r_{1,\mathrm{u}} \\ & r_{2,\mathrm{d}} & 1 \end{bmatrix}, \quad (5)$$

for $K = 2$, which are tridiagonal, and $\widehat{\mathbf{Z}}$ is defined as $\mathbf{Z}$ but with off-diagonal entries $\hat{r}_{k,\mathrm{d}} = r_{k,\mathrm{d}}(1+\varepsilon_{k,\mathrm{d}})$ and $\hat{r}_{k,\mathrm{u}} = r_{k,\mathrm{u}}(1+\varepsilon_{k,\mathrm{u}})$. Diagonal entries of $\mathbf{T}$ are $t_k = c_{k,\mathrm{d}}\Delta_{k,\mathrm{d}}\epsilon_{k,\mathrm{d}} + c_{k,\mathrm{u}}\Delta_{k,\mathrm{u}}\epsilon_{k,\mathrm{u}}$, and off-diagonals are $t_{k,\mathrm{d}} = r_{k,\mathrm{d}}\varepsilon_{k,\mathrm{d}} + c_{k,\mathrm{d}}\Delta_{k,\mathrm{d}}\epsilon_{k,\mathrm{d}}r_{k,\mathrm{d}}$ and $t_{k,\mathrm{u}} = r_{k,\mathrm{u}}\varepsilon_{k,\mathrm{u}} + c_{k,\mathrm{u}}\Delta_{k,\mathrm{u}}\epsilon_{k,\mathrm{u}}r_{k,\mathrm{u}}$, where $\Delta_{k,\mathrm{d}}$ captures the eigenvector misalignment between $\mathbf{L}_{k,\mathrm{d}}$ and perturbation $\mathbf{E}_{k,\mathrm{d}}$ with a factor $\sqrt{n_k}$, and likewise for $\Delta_{k,\mathrm{u}}$.

This result bounds the outputs of a SCCNN on all simplicial levels, showing they are stable to small perturbations on the strengths of simplicial adjacencies and incidences. Specifically, we make two observations from the seemingly complicated expression. 1) The stability bound depends on i) the degree of perturbations including their magnitude $\epsilon$ and $\varepsilon$, and eigenspace misalignment $\Delta$, ii) the integral Lipschitz properties of SCFs, and iii) the degree of projections $r$. 2) The stability of $k$-output depends on factors of not only $k$-simplices, but also simplices of adjacent orders due to inter-simplicial couplings. When $L = 1$, node output bound $d_0$ depends on factors in the node space, as well as the edge space factored by the projection degree, and vice versa for edge output. As the layer deepens, this mutual dependence expands further. When $L = 2$, factors in the triangle space also affect the stability of node output $d_0$, and vice versa for triangle output, as observed in Fig. 1c.

More importantly, this stability provides practical implications for learning on SCs. While accounting for inter-simplicial couplings may be beneficial, it does not help with the stability of SCCNNs when the number of layers increases due to the mutual dependence between different outputs. Thus, to maintain the expressive power, higher-order SCFs can be used in exchange for shallow layers. This does not harm the stability because, first, the components of high-frequency can be spread over the low frequency due to the nonlinearity where the spectral responses are more selective without losing the stability; and second, higher-order SCFs are easier to be learned with smaller integral Lipschitz constants than lower-order ones, thus, better stability. The latter can be easily seen by comparing one-order and two-order cases. We also experimentally show this in Fig. 4.

# 6 Experiments

**Synthetic.** We first illustrate the evolution of Dirichlet energies of outputs on nodes, edges and triangles of a SC of order two by numbers of simplicial shifting layers with $\sigma = \tanh$. The inputs on them are randomly sampled from $\mathcal{U}([-5,5])$. Fig. 2 shows simply generalizing GCN on SCs could lead to oversmoothing on simplices of all orders. However, uncoupling the lower and upper parts of $\mathbf{L}_1$ by setting, e.g., $\gamma = 2$ could mitigate the oversmoothing on edges. Lastly, the inter-simplicial coupling could almost prevent the oversmoothing.

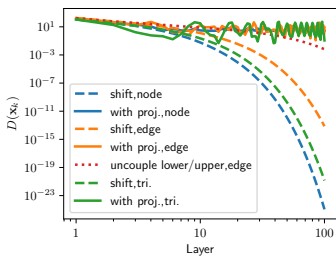

Figure 2

**Foreign currency exchange.** In forex problems, for any currencies $i, j, k$, the *arbitray-free* condition of a fair market reads as $r^{i/j}r^{j/k} = r^{i/k}$ with the exchange rate $r^{i/j}$ between $i$ and $j$. That is, the exchange path $i \to j \to k$ provides no profit or loss over a direct exchange $i \to k$. By modeling the forex as a SC of order two and the exchange rates as edge flows $[\mathbf{x}_1]_{[i,j]} = \log(r^{i/j})$, this condition translates as $\mathbf{x}_1$ is curl-free, i.e., $[\mathbf{x}_1]_{[i,j]} + [\mathbf{x}_1]_{[j,k]} - [\mathbf{x}_1]_{[i,k]} = 0$ in any triangle $[i,j,k]$ [24]. Here we consider a real-world forex market from [50] at three timestamps, which contains certain degree of arbitrage. We artificially added some random noise and "curl noise" (only in the curl space) to this market, in which we aim to recover the forex rates. We also randomly masked 50% of the rates, where we aim to interpolate the market such that it is arbitrage-free. Three settings create three types of learning needs. To evaluate the performance, we measure both normalized mse and total arbitrage (total curl), both equally important for the goal of *creating a fair market by small price fluctuations*.

From Table 1, we make the following observations. 1) MPSN [17] fails at this task: although it can reduce nmse, it outputs unfair rates with large arbitrage, which is against the forex principle, because it is not Hodge-aware, unable to capture the arbitrage-free property with small amount of data. 2) SNN [15] fails too: as discussed in Section 4, it restricts the gradient and curl spaces to be always learned in the same fashion, unable to meet the need of disjoint learning of this task in two

Table 1: Forex results (nmse|total arbitrage).

| Methods | Random Noise | Curl Noise | Interpolation |
|---|---|---|---|
| Input | $.119_{\pm.004}$\|$29.19_{\pm.874}$ | $.552_{\pm.027}$\|$122.4_{\pm5.90}$ | $.717_{\pm.030}$\|$106.4_{\pm.902}$ |
| Baseline | $.036_{\pm.005}$\|$2.29_{\pm.079}$ | $.050_{\pm.002}$\|$11.12_{\pm.537}$ | $.534_{\pm.043}$\|$9.67_{\pm.082}$ |
| SNN [15] | $.110_{\pm.005}$\|$23.24_{\pm1.03}$ | $.446_{\pm.017}$\|$86.95_{\pm2.20}$ | $.702_{\pm.033}$\|$104.74_{\pm1.04}$ |
| PSNN [16] | $.008_{\pm.001}$\|$.984_{\pm.170}$ | $.000_{\pm.000}$\|$.000_{\pm.000}$ | $.009_{\pm.001}$\|$1.13_{\pm.329}$ |
| MPSN [17] | $.039_{\pm.004}$\|$7.74_{\pm0.88}$ | $.076_{\pm.012}$\|$14.92_{\pm2.49}$ | $.117_{\pm.063}$\|$23.15_{\pm11.7}$ |
| SCCNN, id | $.027_{\pm.005}$\|$.000_{\pm.000}$ | $.000_{\pm.000}$\|$.000_{\pm.000}$ | $.265_{\pm.036}$\|$.000_{\pm.000}$ |
| SCCNN, tanh | $\mathbf{.002_{\pm.000}}$\|$\mathbf{.325_{\pm.082}}$ | $.000_{\pm.000}$\|$.003_{\pm.003}$ | $\mathbf{.003_{\pm.002}}$\|$\mathbf{.279_{\pm.151}}$ |

Table 2: Simplex prediction.

| Methods | 2-simplex | 3-simplex |
|---|---|---|
| Mean [2] | 62.8±2.7 | 63.6±1.6 |
| MLP | 68.5±1.6 | 69.0±2.2 |
| GNN [51] | 93.9±1.0 | 96.6±0.5 |
| SNN [15] | 92.0±1.8 | 95.1±1.2 |
| PSNN [16] | 95.6±1.3 | 98.1±0.5 |
| SCNN [19] | 96.5±1.5 | 98.3±0.4 |
| Bunch [14] | 98.3±0.5 | 98.5±0.5 |
| MPSN [17] | 98.1±0.5 | 99.2±0.3 |
| **SCCNN** | **98.7±0.5** | **99.4±0.3** |

Table 3: Ablation study.

| Missing | 2-Simplex | Param. |
|---|---|---|
| — | 98.7±0.5 | $L=2$ |
| Edge-to-Node | 93.9±1.0 | $L=5$ |
| Node-to-Node | 98.7±0.4 | $L=4$ |
| Edge-to-Edge | 98.5±1.0 | $L=3$ |
| Node-to-Edge | 98.8±0.3 | $L=4$ |
| Node input | 98.2±0.5 | $T=4$ |
| Edge input | 98.1±0.4 | $T=3$ |

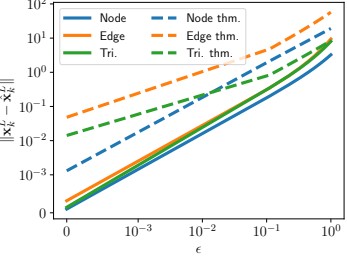

Figure 3: Stability bound.

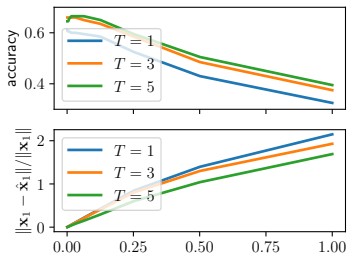

Figure 4: Stability as $T$ increases.

spaces. 3) PSNN [16] can reconstruct relatively fair forex rates with small nmse. In the curl noise case, the reconstruction is perfect, while in the other two cases, the nmse and arbitrage are three times larger than SCCNN due to its limited linear learning responses. 4) SCCNN performs the best in both reducing the total error and the total arbitrage. We also notice that with $\sigma = \mathrm{id}$, the arbitrage-free rule is fully learned by SCCNN. However, it has relatively larger errors in the random and interpolation cases due to its limited linear expressive power. With $\sigma = \tanh$, SCCNN can tackle these more challenging cases, finding a good compromise between overall error and data characteristics.

**Simplex Prediction.** We then test SCCNN on simplex prediction task which is an extension of link prediction in graphs [52]. Our approach is to first learn the features of lower-order simplices and then use an MLP to identify if a simplex is closed or open. We built a SC as [15] on a coauthorship dataset [53] where nodes are authors and collaborations of $k$-authors are $(k-1)$-simplices. The input simplicial data is the number of citations, e.g., $\mathbf{x}_1$ and $\mathbf{x}_2$ are those of dyadic and triadic collaborations, which does not present explicit properties like forex rates. Thus, 2-simplex (3-simplex) prediction amounts to predict triadic (tetradic) collaborations. From the AUC results in Table 2, we make three observations. 1) SCCNN, MPSN and Bunch [14] methods outperform the rest due to the inter-simplicial couplings. 2) Uncoupling the lower and upper parts in $\mathbf{L}_k$ imrpoves the feature learning (SCNN [19] better than SNN). 3) Higher-order convolution further improves the prediction (SCCNN better than PSNN, SCCNN better than Bunch). Note that MPSN has three times more parameters than SCCNN under the settings of the best results.

**Ablation study.** Table 3 reports the results of SCCNN when certain simplicial relation is missing, which helps understand their roles. When not considering the edge-to-node incidence, it (when using node features) is equivalent to GNN with poor performance. When removing other adjacencies or incidences, the best performance remains but with an increase of model complexity, more layers required. This, however, is not preferred, because the stability decreases as the model deepens and becomes influenced by factors in other simplicial space, as shown in Fig. 1c. We also considered the case with limited input, e.g., when the input on nodes or on edges is missing. The best performance of SCCNN only slightly drops with an increase of convolution order, compared to before $T = 2$.

**How tight is the stability bound?** We consider the perturbations which relatively shift the eigenvalues of Hodge Laplacians and the singular values of projection matrices by $\epsilon$. We compare the bound in Eq. (5) with experimental distance on each simplex level. Fig. 3 shows the bound becomes tighter as perturbation increases.

Table 4: Trajectory prediction.

| Methods | Synthetic | Ocean drifts |
|---|---|---|
| SNN [15] | 65.5±2.4 | 52.5±6.0 |
| PSNN [16] | 63.1±3.1 | 49.0±8.0 |
| SCNN [19] | **67.7±1.7** | 53.0±7.8 |
| Bunch [14] | 62.3±4.0 | 46.0±6.2 |
| SCCNN | 65.2±4.1 | **54.5±7.9** |

**Trajectory prediction.** We lastly test on predicting trajectories in a synthetic SC and of ocean drifters from [41], introduced by [16]. From Table 4 we first observe SCCNN and Bunch with inter-simplicial couplings do not perform better than those without. This is because zero inputs are applied on nodes and triangles [16], which makes inter-couplings inconsequential. Secondly, using higher-order convolutions improves the

average accuracy in both datasets (SCNN better than PSNN on average, SCCNN better than Bunch). Note that the prediction here aims to find a candidate from the neighborhood of end node, which depends on the node degree. Since the average node degree of the synthetic SC is $5.24$ and that in ocean drifter data is $4.81$, a random guess has around $20\%$ accuracy. The high standard derivations could come from the limited ocean drifter dataset.

**Convolution orders on stability.** We also show that NNs with higher-order SCFs have more potential to learn better integral Lipschitz properties, thus, better stability. We consider SCNNs [19] with orders $T_\mathrm{d} = T_\mathrm{u} = 1, 3, 5$ and train them with a regularizer to reduce the integral Lipschitz constants. As shown in Fig. 4, the higher-order case has a smaller distance (better stability) between the outputs without and with perturbations, with consistent better accuracy, comapred to the lower-order case.

# 7    Related Work, Discussion and Conclusion

Related work mainly concerns learning methods on SCs. [13] first used $\mathbf{L}_{1,\mathrm{d}}$ to build NNs on edges in a graph setting without the upper edge adjacency. [15] then generalized convolutional GNNs [32, 51] to simplices by using the Hodge Laplacian. [16, 19] instead uncoupled the lower and upper Laplacians to perform one- and multi-order convolutions, to which [42, 38, 54] added attention schemes. [55] considered a varaint of [16] to identity topological holes and [18] combined shifting on nodes and edges for link prediction. Above works learned within a simplicial level and did not consider the incidence relations (inter-simplicial couplings) in SCs, which was included by [14, 20]. These works considered convolutional-type methods, which can be subsumed by SCCNN. Meanwhile, [17, 21] generalized the message passing on graphs [22] to SCs, relying on both adjacencies and incidences. Most of these works focused on extending GNNs to SCs by varying the information propagation on SCs without many theoretical insights into their components. Among them, [16] discussed the equivariance of PSNN to permutation and orientation, which SCCNN admits as well. [17] studied the messgae-passing on SCs in terms of WL test of SCs built by completing cliques in a graph. The more closely related work [19] gave only a spectral formulation based on SCFs.

**Discussion.** In our opinion, the advantage of using SCs is not only about them being able to model higher-order network structure, but also support simplicial data, which can be both human-generated data like coauthorship, and physical data like flow-typed data. This is why we approcahed the analysis from the perspectives of both simplicial structures and the simplicial data, i.e., the Hodge theory and spectral simplicial theory [23, 9, 26–28, 30, 56]. We provided deeper insights into why three principles are needed and how they can guide the effective and rational learning from simplicial data. As what we practically found, in experiments where data exhibits properties characterized by the Hodge decomposition, SCCNN performs well due to the Hodge-awareness while non-Hodge-aware learners can fail at giving rational results. In cases where data does not possess such properties, SCCNN has better or comparable performance than the ones which violate or do not respect the three principles. This also shows the advantages of SCCNN, especially when data has certain properties.

Concurrently, there are works on more general cell complexes, e.g., [57–61], where 2-cells inlcude not only triangles, but also general polygon faces. We focus on SCs because a regular CW complex can be subdivided into a SC [62, 29] to which the analysis in this paper applies, or we can generalize our analysis by allowing $\mathbf{B}_2$ to include 2-cells. This is however informal and does not exploit the power of cell complexes, which lies on cellular sheaves, as studied in [63, 64].

**Limitation.** A major limitation of our method is that it cannot learn differently from features at the frequencies of the same type and the same value. For instance, harmonic features are learned in a same fashion because they all have zero frequency. This is however common in convolutional type learning methods on both graphs and SCs. Also, our stability analysis concerns the perturbations on the connection strengths and did not consider the case where simplices join or disappear. Both of them can be interesting future directions, together with more physical-based data applications.

**Conclusion.** We proposed three principles for convolutional learning on SCs, summarized in a general architecture, SCCNN. Our analysis showed this architecture, guided by the three principles, demonstrates an awareness of the Hodge decomposition and performs rational, effective and expressive learning from simplicial data. Furthermore, our study reveals that SCCNN exhibits stability and robustness against perturbations in the strengths of simplicial connections. Experimental results validate the benefits of respecting the three principles and the Hodge-awareness. Overall, our work establishes a solid fundation for learning on SCs, highlighting the importance of the Hodge theory.

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
