# OpenReview forum: "Hodge-Aware Learning on Simplicial Complexes"
_NeurIPS.cc/2023/Conference — Submitted to NeurIPS 2023_

### Official Review · Reviewer_shzQ · 2023-07-03

**Soundness:** 3 good
**Presentation:** 3 good
**Contribution:** 2 fair
**Rating:** 4
**Confidence:** 3

**Summary:**

In this paper, the authors first use Dirichlet energy minimizations on simplicial complexes (SCs) to interpret their effects on mitigating the simplicial oversmoothing. Then, through the lens of spectral simplicial theory, they show the three principles promote the Hodge-aware learning of this architecture, in the sense that the three Hodge subspaces are invariant under its learnable functions and the learning in two nontrivial subspaces are independent and expressive. Moreover,  the authors prove it is stable against small perturbations on the strengths of simplicial connections, and show how three principles can affect the stability. Lastly, we validate our findings on different simplicial tasks, including recovering foreign currency exchange (forex) rates, predicting triadic and tetradic collaborations, and trajectories.

**Strengths:**

1. The paper is well-written. The introduction and background give a nice overview and motivation for the problem.
2. The problem of simplicial complexes learning is really interesting and in many aspects understudied.
3. In the simplex prediction, the proposed SCCNN achieves promising empirical performance.
4. The proposed SCCNN is supported by a theoretical analysis.


**Weaknesses:**

1. Limited applications and examples. It would be interesting to see more applications of the proposed SCCNN over widely used datasets (e.g., citation networks - Cora, CiteSeer, PubMed) for node and graph classification tasks. Moreover, although SCCNN achieves promising performance compared with SC-based model, can the authors compare it with other the state-of-the-art graph neural network (GNN)-based models?
2. Can the authors provide the running time of SCCNN and compare it with the state-of-the-art baselines?
3. How to select the dimension of $k$-simplex in the SCCNN model?

**Questions:**

See comments and questions in Weaknesses.

**Limitations:**

In general, I think this is a good paper with tackling a well motivated task. It would be helpful that this paper explores more datasets and compares with more advanced graph neural network-based models.

---

> ### Author Rebuttal · Authors · 2023-08-04
>
> We appreciate the reviewer's feedback. Please find below our replies.
> > @ #1: applying SCCNN for node/graph classifications, comparing with sota GNN
>
> We understand your interest in how SCCNN performs in node/graph classifications.
> These are important problems but do not fit the goal of this paper. We actually do not advocate using any simplicial-based learning for graph-based tasks as the domain is different but rather using them for topology-based tasks involved with simplicial data where GNNs cannot be applied. We will elaborate on this below but please allow us first to do a recap of the paper's goal.
>
> _Recap: What is our goal?_ Building a principled learning method for __simplicial signals__ with theoretical guarantees, drawing its foundation from the Hodge theorem.
>
> _Recap: Why this goal?_ The Hodge theorem, which characterizes simplicial signals in three subspaces, has not been considered in existing learning methods for simplicial signals.
>
> _Recap: What is not the goal_? To improve the state-of-the-art on graph tasks (node/graph classification). These are not our goals, though important.
>
> Why those are not the goal?
>
> Node classification is commonly done by learning node representations without engaging higher-order simplicial data. Thus, it is not a task focused by simplicial network solutions [13-20]. In fact, using SCCNN is equivalent to the standard GNN, and MPSN [17] is equivalent to a graph isomorphism network (GIN) (when using node features).
>
> Graph classification.
>
> 1. _Complexity issue_. While testing SCCNN on graph classification is possible, one needs to lift a graph to a clique complex, as done by [17] and [61], in which the number of $k$-simplices $n_k$ is generally large with the worst case $n_k=\mathcal{O}(n^{k+1})$, with the number of nodes $n$. This was discussed in [17,p5] and [61,p10].
>
> 2. MPSN [17] and [61] achieved comparable performance by including the simplicial or cell structures in graph classifications. These works, among others, revealed the __Topology__ asset of simplicial complexes (SCs) in modeling higher-order simplicial structures.
> However, we have a focus on the power of SCs in supporting simplicial signals defined on higher-order simplicial structures (which _GNN models are not designed for_), e.g., edge flows, triangle signals, etc. This exploits both __Topology__ and __Algebra__ assets of SCs. Such signals are well motivated, discussed in L27-29, 42-47, 393, and considered in our experiments. _The Hodge theorem provides as a natural tool to characterize these signals, which is however not studied in existing works_.
>
> These motivate our paper, a _principled Hodge-aware learning_ method for simplicial signals. Applying SCCNN to graph classification as in [17,61] is possible and interesting, but it does not exploit the asset of SCs to support simplicial signals, nor does it help concentrate the core of the paper. It will hide our main message, which is to show how the three principles can aid learning simplicial signals in the SC domain.
> While applying more advanced GNN models to simplex prediction is possible, this would hinder one to fairly conclude from the comparison between graph and simplicial convolutions in this task, as they may introduce new experimental variables from their advanced techniques, compared to the standard convolutional one used by us.
>
> Instead, we focus on the exploratory tasks, forex learning and simplex prediction, as well as trajectory prediction following [16], which all involve with edge flows and triangle signals, demonstrating potential applications of SCs. We did an extensive experimental study and a thorough comparison with existing simplicial methods, which are more important for the paper. They sufficiently and necessarily validate our theoretical claims on the principles behind SCCNN, the Hodge-awareness and the stability results.
>
> We hope this helps the reviewer to see that the tasks on SCs are not limited to graph tasks, the motivation for our simplicial-based tasks, and more importantly, the broader contributions behind SCCNN for learning simplicial signals.
>
> > @ #2: about running time
>
> Yes, certainly. We analyzed the computational complexity of SCCNN in L134 and Appendix B.2. Here we report the number of parameters and the running time in seconds per epoch of SCCNN for 2-simplex prediction on one NVIDIA Quadro K2200 with 4GB memory, compared with the two best alternatives.
>
> |Method|#params.|Running time|
> |-|-|-|
> |SCCNN|24288|0.073|
> |Bunch|21728|0.140|
> |MPSN|84256| 0.028|
>
> MPSN, compared to convolutional methods, has three times more parameters, analogous to the comparison between message-passing and graph convolutional NNs. The running time of MPSN is smaller.
>
> We also report the running time as the layers $L$ and convolution orders $T$ increase.
> |Hyperparams.|$T=2$|$T=5$|
> |-|-|-|
> |$L=2$|0.073|0.082|
> |$L=3$|0.110|0.130|
> |$L=5$|0.192|0.237|
>
> We see that the efficiency of using a higher-order convolution with $T=5$ does not drastically increase compared to a lower-order one with $T=2$. However, as $L$ increases, the efficiency can significantly decrease. This also supports the use of higher-order convolutions with a small number of layers, which is beneficial for stability as studied in Section 4 (L296).
>
> > @ #3: about the dimension of $k$-simplex
>
> We assume the question is about the number of $k$-simplices, $n_k$, which is given by the underlying SC, e.g., $n_1$ is the number of edges, $n_2$ is the number of 2-simplices, which are filled triangles, different from open triangles (any three pairwise connected nodes).
> In general, $n_2$ is much smaller than $\mathcal{O}(n^3)$, in analogy to $n_1\ll\mathcal{O}(n^2)$ in a graph, with $n$ the number of vertices. In practice, an SC can be constructed via well-studied algorithms in a well-defined metric space, e.g., VR complex, Cech complex. This often concerns the field of topological data analysis.
>
> We hope our replies properly address your questions.

---

> > ### Comment · Reviewer_shzQ · 2023-08-17
> > **Official Comment by Reviewer shzQ**
> >
> > Thanks for the detailed response from the authors. Since the paper still requires non-trivial efforts to make it above the NeurIPS bar, I will keep my score the same.

---

> > > ### Author Response · Authors · 2023-08-17
> > >
> > > Thanks for checking our answer. We would appreciate if the reviewer could be specific about "the paper still requires non-trivial efforts to make it above the NeurIPS bar". We believe that we have effectively addressed all three weaknesses that were initially pointed out.
> > >
> > > It seems that this might pertain to Weakness #1, where the reviewer asked for GNN tasks (node/graph classification) and a comparison with advanced GNNs.
> > > In our rebuttal,  we provided reasoning supported by both theoretical foundations and existing literature.
> > >
> > > We would like to draw a parallel here: _To some extent_, requesting GNN tasks and comparisons with advanced GNN methods in a _simplicial neural network_ paper could be compared to requesting CNN tasks and comparisons with advanced CNN methods in a GNN paper where 2D grids can be viewed as special cases of graphs.
> > >
> > > While the reviewer's suggestion is intriguing, it does not directly align with the core claims of our paper, which aims to promote learning simplicial signals based on the Hodge decomposition. Instead,
> > > 1. we focused on tasks that involve learning simplicial signals (on edges and triangles): two exploratory tasks (forex learning, simplex prediction) on real data and one task (trajectory prediction) following [16] on both synthetic and real data.
> > > 2. we have conducted thorough comparisons with other simplicial-based methods.
> > > 3. we conducted extensive stability study and ablation study (Appendix F) to support the claims of the paper.
> > >
> > > Considering the limited space within the paper, which is dedicated to _learning on simplicial complexes_, we believe that simplicial tasks are _both necessary and sufficient_, even though graph tasks are interesting. While we understand that readers with a GNN background may be interested in graph tasks, focusing on graph tasks in our paper does not convey the core the Hodge-aware learning, nor the advantage of using simplicial complexes.
> > >
> > > Thus, we believe the numerical experiments of the paper corroborate the theory and highlight its main contributions, while comparisons in graph-based tasks would not add many insights to the paper.

---

### Official Review · Reviewer_XgBj · 2023-07-07

**Soundness:** 2 fair
**Presentation:** 3 good
**Contribution:** 2 fair
**Rating:** 4
**Confidence:** 3

**Summary:**

The authors identified the limitations of non-Hodge aware learners on simplicial complex (SC) data and proposed a convolutional structure that 1) decomposes the upper and lower k-Laplacian and 2) takes the inter-simplicial couplings into account. The paper has presented a justification for the performance by analyzing the Dirichlet energy and oversmoothing. Additionally, they provided theoretical perturbations bound to study the robustness of the proposed convolution layer. The claims are supported by experiments on synthetic and real SC datasets.

**Strengths:**

1. [Originality] Incorporate the well-known Hodge theorem into the learning task on simplicial complex. The Hodge theorem provides a good intuition and explanation for the learning of the simplicial signal on SC.
1. [Quality] Empirical examples on the synthetic datasets on Dirichlet energy and stability bound to support the theoretical claims.
1. [Clarity] Great overview of the simplicial complex and Hodge decomposition. The authors also provided a motivation/justification for why the proposed layer works with Dirichlet energy minimization.
1. [Significance] Being able to learn the simplicial signal in different Hodge subspaces is an important task.

**Weaknesses:**

1. If this framework needs to be applied to graph only having edges (i.e., SC of order 1), one usually can apply something like clique-complex (or any other methods to fill in the Simplicios) from that graph. In this case, $n_k$ is generally large (worst case $n_k = \mathcal O(n^k)$), resulting in a huge $L_k$ matrix. How practical is it to use the proposed method under this scenario?
1. Can you provide a definition/discussion or citation of Hodge-aware? It is not clear to me where it is defined throughout the manuscript. I can get some high-level ideas by reading Theorem 7, but I think it would be nice if you could explicitly call it out at the beginning (e.g., in introduction or background).
1. The discussion for preventing “over-smoothing” in Section 3 is great, it provides some high-level motivations of the choices you made. However, I am not sure if that it can support the claims.  Specifically, to really prevent “over-smoothing” of the Dirichlet energy, shouldn’t we bound the $D(x_k^{\ell+1})$ in other way around, i.e., with a lower bound rather than an upper bound? If we can show that $D(x_k^{\ell+1})$ can be lower bounded, the claim can be more convincing.
1. Consider adding some high-level intuition on what harmonic flow is using the edge space example (e.g., flow cycling around global topoplogical structures); this will give readers having no background in Hodge decomposition a better understanding of what a “harmonic flow” is.
1. [Typo] L186 there is typo/grammatical issue, do you mean “$\tilde{h}_k = \text{diag}(...)$ is the frequency response of $\mathbb H_k$”?


**Questions:**

1. Related to Weakness #2, why the Hodge Laplacian smoothing [31] not Hodge-aware? I think it can also learn from the different subspaces of the Hodge Laplacian, just not independently. Maybe adding some definition/citation as per #2 will clarify it a bit.
1. [Minor language usage suggestion] Consider rewrite L24-L25 to improve clarity; for instance, you might rewrite it as something like “A SC can be informally viewed as an extension of a graph. For example, one of the simplest SC (SC_2) can be constructed from a graph by inducing some triangles over the edge set.”.
1. [Minor language usage suggestion] There is an extra e.g., in L27
1. [Minor language usage suggestion] Consider breaking L27-29 into multiple sentences to improve clarity.
1. [Minor notation issue] I would consider changing the notation of the $\mathbf B$ matrix in L259 to reduce confusion.

**Limitations:**

The paper discusses some of its limitations and requirements/assumptions. No significant social impact is identified from this work.

---

> ### Author Rebuttal · Authors · 2023-08-04
>
> We appreciate the reviewer's careful and critical feedback. Here we address the detailed concerns.
> > @ Weakness #1: about complexity of $n_k$
>
> It is not practical to use SCCNN under this scenario due to the high complexity of $(k+1)$-cliques ($k$-simplices), $n_{k}=\mathcal{O}(n^{k+1})$. This also holds for other learning methods on simplicial complexes (SCs). [17] (and [61]) applied message-passing on SCs (and cell complexes) lifted from the underlying graphs in this way to perform graph classification. Both [17,p5] and [61,p10] discussed this complexity limitation. This is one of the reasons why we did not perform graph classification following [17].
>
> Instead, we motivated and emphasized the asset of using SCs in supporting simplicial signals (Lines 27-29, 42-47, 393-403, and our experiments), where the underlying SCs are given with the sparsity that $n_k\ll\mathcal{O}(n^{k+1})$, in analogy to the number of edges follows $n_1\ll\mathcal{O}(n^2)$ in real-world graphs. This is the case in our four experiments (Appendix F). In terms of the construction of SCs, this however concerns the field of topological data analysis. For example, given a metric space, one can use Delaunay triangulation, or construct Rips or Cech complex, which comes with use-defined sparsity.
>
> > @ Weakness #2: about Hodge-aware...
>
> The definition of Hodge-aware is given by us, composed of three theoretically analyzable properties.
> In Theorem 7, we gave its three thumbnail properties. While the independent and expressive learning follows from the previous discussion (eq.4 and Theorem 6), we omitted the formulations for the Hodge subspaces being $H_k$-invariant. This may appear high-level.
>
> Please refer to our __Global Response__ where we gave a more formal definition and lengthier discussions. For a better exposition, we also added an illustration in the __attached PDF__.
>
> _Our action_: We will define invariant subspace in Background and include lengthier discussion and the PDF figure in Section 4.
> In Introduction [Lines 40-50], we motivated the need of effective learning in the characteristic Hodge subspaces, thus, Hodge-awareness.
>
> About possible literatures, the invariance of three Hodge subspaces has appeared in _E. Nijholt and L. DeVille, Dynamical systems defined on simplicial complexes: symmetries, conjugacies, and invariant subspaces, Chaos, 2022._
> Also, we refer to _Beauzamy. Introduction to operator theory and invariant subspaces. Elsevier, 1988._ for the notion of invariant subspace of an operator, which is commonly used to understand a complicated operator.
>
> Consider an operator $T:V\to V$ defined on some big space $V$. Assume $V$ admits a direct sum $V=W_1\oplus\dots\oplus W_k$, where each subspace $W_i$ is $T$-invariant, i.e., $Tw_i\in W_i$ for $w_i\in W_i$. Then, we can understand the operators $T|_{W_i}:W_i\to W_i$ (the restriction on $W_i$), which are defined on smaller spaces. If the $W_i$ is not $T$-invariant, then $T$ would "mix $W_i$ with $W_j$" for $j\neq i$. We cannot exploit the direct sum.
>
> In our case, $H_k$ corresponds the $T$ above, to which the Hodge subspaces are invariant. While having this invariant property reduces the learning space, we also want to have independent learning in different Hodge subspaces, as they characterize different parts of the signal. Lastly, we want the operator to be expressive enough so to approximate any operators in each subspace.
> This motivates the three properties for the Hodge-aware learning.
>
> > @ Weakness #3: about the lower bound $D(x_k)$...
>
> Yes, we agree. Notice that the objective eq.(3) can have a zero minimum. This corresponds to a lower bound of Dirichlet energy, $D(x_k)\geq\lVert x_{k-1}\rVert^2+\lVert x_{k+1}\rVert^2$, where inter-simplicial coupling acts as energy sources.
> In fact, our discussion in [Lines 163-164], that the optimal solution is in the images of $B_{k+1}$ and $B_k$, implicitly gave the lower bound. But we will replace the upper bound by this lower bound.
>
> > @ Weakness #4: illustrating the harmonic flow
>
> We agree. We actually have an illustration Figure A.1 in Appendix A. If possible, we will move it to page 2.
>
> > @ Weakness #5: about typo...
>
> $\tilde{h}\_k$ is the frequency response of $H_k$. It can be written w.r.t. the three types of frequencies in detail, i.e., $\tilde{h}\_{k,H},\tilde{h}\_{k,G},\tilde{h}\_{k,C}$ as in the RHS of eq.(4).
>
> > @ Question #1: why Hodge Laplacian smoothing is not Hodge-aware
>
> First, your premise is correct. While it can learn from different subspaces, the learning in the gradient and curl spaces is not independent, which is required by Hodge-aware learning.
>
> This independent learning is reflected as two independent spectral responses, $\tilde{h}\_{G}(\lambda)$ for gradient space and $\tilde{h}\_{C}(\lambda)$ for curl space, which are learned by two sets of weights (eq.(4)). (Topologically, this corresponds to the independent convolution via the lower and upper adjacencies.)
> Instead, a non-independent learner has a common spectral response $\tilde{h}(\lambda)$ for two subspaces. An intuitive illustration can be found in Figure 1(a) or in the attached PDF.
>
> 1. This is critical at a $\lambda$ that is both gradient and curl frequency. Consider a signal only in the gradient space with some noise in the curl space. A non-independent learner can either remove or preserve both signal and noise at $\lambda$ while an independent learner learns differently. This is why SNN [15] failed in forex where exchange rates should not be in the curl space [Line 329].
> 2. This increases the learning expressive power due to the increase of degree of freedom. We observe the performance increase in simplex and trajectory predictions, e.g., SNN vs SCNN, and Bunch vs SCCNN.
>
> > @ Questions 2-5: about minor suggestions
>
> We made corresponding changes in the paper and appreciate the reviewer's help.
>
> > @ Our final words
>
> We hope our answers, together with Global Response and the PDF, properly address your questions.

---

> > ### Author Response · Authors · 2023-08-18
> >
> > Dear Reviewer,
> >
> > As the discussion phase ends soon, we would like to kindly ask if our answers addressed your questions about weaknesses #1-3 and question #1. Especially, in our rebuttal,
> > - for weakness #2 about _Hodge-aware_, we gave an example to show why the three properties are introduced in the Hodge-aware learning. Consider a general learning operator $T:V\to V$ where $V$ admits a direct sum decomposition, $V=W_1\oplus\dots\oplus W_k$. This generalizes our setting where our learning operators are mappings between simplicial signal spaces, which admit the Hodge decomposition. The three properties of Hodge-aware learning, respectively, correspond to
> > >1. each $W_i$ is an invariant subspace of $T$ (in Hodge-aware: _each Hodge space is an invariant subspace of the learning operators $H_k$_),
> > >2. the restriction of $T$ on each subspace, $T|_{W_i}:W_i\to W_i$, is controlled by independent learning parameters (in Hodge-aware: _we need independent learning in the gradient and curl subspaces_), and,
> >  >3. $T$ being expressive such that its restriction on each subspace, $T|_{W_i}$, can approximate any mappings between $W_i$s (in Hodge-aware: _learning operators of SCCNN are expressive, as studied in Thm. 6_).
> >
> >     Also, we gave a lengthier discussion in the __Global Response__, together with an illustration attached in the __PDF__. Please refer to that for more details.
> > - for weakness #3, we will explicitly add the lower bound.
> > - for the rest language usage suggestions, we will incorporate them.
> >
> > Once again, we appreciate your input in reviewing our paper, specifically, your detailed questions and suggestions on improving the paper.

---

### Official Review · Reviewer_d7bL · 2023-07-10

**Soundness:** 2 fair
**Presentation:** 1 poor
**Contribution:** 2 fair
**Rating:** 2
**Confidence:** 3

**Summary:**

This paper aims to propose a convolutional architecture which incorporates the Hodge theory. Specifically, the proposed architecture incorporates the following three properties: uncoupling the lower and upper simplicial adjacencies, accounting for the inter-simplicial couplings, and performing higher-order convolutions.




**Strengths:**

The paper claims a new architecture that incorporates the following three properties: uncoupling the lower and upper simplicial adjacencies, accounting for the inter-simplicial couplings, and performing higher-order convolutions, but the clear differences with respect to existing simplicial neural models are unclear.

**Weaknesses:**

- Errors: The paper is very poorly written. The typographic, punctuation, and grammatical errors throughout the paper make it hard to follow. Lines 21, 25, 31, 32, and 37 (in the first page of the paper) are some examples of lines that containing such errors. Many abbreviations are used before defining them. For example, NN (line 30), SCCNN (line 51), MLP (line 65), SCF (line 264) etc., were never defined. Many variables, like \mathcal{V} in line 70 and most of the variables in eq. (4), and terms, like alternating map in line 87, were never introduced.
- Contributions: The contributions of the paper are unclear. In line 104, for example, the authors say, “we inherit the names of three edge subspaces to general k-simplices”, while the Hodge theory is already in place for simplices of all dimensions. The main contribution of the paper, which is supposed to be an architecture that incorporates the Hodge theory, is also not clearly presented.



**Questions:**

- It is not clear if Section 3 is about the proposed model or an explanation of the existing models.  The expression in eq. (1) is a general expression for the existing SNN models, which is also what the authors say in line 119. Given this, it is not clear what the contribution of the paper is.  Even the three properties in lines 119-125 that the paper claims to be present in the proposed model, I believe, are possessed also by MPSN. In the section named “From convolutional to Hodge-aware”, where the authors compare the proposed model to the existing models, the comparison with MPSN is missing. Furthermore, the motivation for defining Dirichlet energy the way it is defined in Definition 2 is not clear. Should it not be || (B_k + B_{k+1}^T)x_k ||^2 if it was a direct extension from graphs? If it was defined as in Definition 2 in some work earlier, the reference to the work should be provided.

- The authors claim that the good performance of the proposed model is due to the three properties (lines 345-348) that it incorporates. Since MPSN also incorporates the three properties, should it not perform as well as the proposed model?

**Limitations:**

Overall, the paper is very hard to follow. The motivation and contributions of the work are not clear. The errors in the paper make it more difficult to follow

---

> ### Author Rebuttal · Authors · 2023-08-04
>
> We thank the reviewer for reviewing our manuscript and point out some textual and grammar inconsistencies in it.
>
> > @ "Errors:..." in the Weakness
>
> Our apologies from the grammar errors, which were due to continuously shortening and updating the manuscript till minutes before the deadline. We have carefully checked and updated all the aspects you mentioned.
>
> Regarding the acronyms, only two popular ones are not explicitly stated (NN - neural network) and (MLP - multi-layer perceptron). The SCF is defined in Line 84.
>
> Set $\mathcal{V}$ is defined as $\mathcal{V} = \{1,\dots,n_0\}$ in Line 70.
>
> Variables in eq. (4): we will give clear definitions one by one. Thanks for pointing this out as some of them were defined only in the supplement.
>
> Alternating map in Line 87: we immediately gave its antisymmetric property in Lines 87-88, which is what needed in this paper. We will add a formal definition of it as initially we wanted to avoid many mathematical jargons.
>
> > @ "Contributions: "
>
> We are a bit surprised from this comment as in the second last paragraphs of the introduction state: "In this paper, we present ..." claiming the SCCNN as architectural contribution and the last paragraph "Main contributions" claiming the paper contributions revolving on the theoretical and empirical analysis of this architecture. Maybe this is a writing style but listing them, the contributions of this paper are:
>
> c1) It proposed the SCCNN architecture; eq. (1);
>
> c2) It studies its equivariances w.r.t. the topological domain; Sec. 3.1
>
> c3) We show how the intra and inter-couplings in the simplicial complex can mitigate oversmothing; Sec. 3.2
>
> c4) We characterize the expressive power of the SCCNN and shows it performs Hodge-aware learning; Sec. 4
>
> c5) We prove the SCCNN is stable do domain perturbations in Sec. 5. This is the first-ever stability results in the simplicial domain;
>
> c6) We corroborate all these theoretical findings with experiments and show how they play a role in improving the performance w.r.t. state-of-the-art alternatives; Sec. 6
>
> Regarding "we inherit...", we just state that we will use these names to general simplicial spaces (from the edge space) to ease the exposition. Definitively this is not our contribution and we do not claim so. This is indeed in the Background section.
>
> The similarities and differences w.r.t. earlier works are done in Sec. 7. Moreover, in Lines 216-218 and 228-232, 234-242, the comparisons with other methods, including MPSN, are done in the Hodge-aware sense. We will make all these aspects more explicit and direct.
>
> > @ "Question 1"
>
> - Sec. 3 is about the proposed model (SCCNN). We made this explicit now.
> - Expression (1) is our general architecture. In line 119, we say that it generalises the existing solutions. Our model does generalize the existing convolutional type neural networks on simplicial complexes. But this is only a minor contribution of the paper.
> - Properties in lines 119-125: The MPSN does not satisfy property 3) as each layer performs one message passing step. Thus, it performs only one-order operations, since it aggregates messages from its direct lower and upper neighbors. A result of this is it requires deep layers to collect information from neighbors far away, compared to higher-order convolution. Moreover, as we studied in the stability, this leads to high instability due to the use to deep layers and mutual negative dependencies between outputs on different levels of simplices.
> The simplicial convolutional networks satisfy 3) but not 1) and 2).
> - Hodge-aware and MPSN: we explicitly discussed that MPSN is not Hodge-aware in Lines 216-218 and 228-232.
> - Dirichlet energy: the definition is $\lVert (B_k+B_{k+1}^\top)x_k \rVert_2^2 = \lVert B_kx_k \rVert_2^2 + \lVert B_{k+1}^\top x_k \rVert_2^2$. This is due to that $B_kB_{k+1}=0$ (line 82). We here expand this step by step.
>  $
>         \lVert (B_k+B_{k+1}^\top)x_k \rVert_2^2
>         = x_k^\top(B_k + B_{k+1}^\top)^\top (B_k + B_{k+1}^\top) x_k
>         = x_k^\top(B_k^\top B_k + B_{k+1}B_{k+1}^\top + B_k^\top B_{k+1}^\top + B_{k+1}B_k) x_k
>         =  x_k^\top (B_k^\top B_k + B_{k+1}B_{k+1}^\top)x_k
>         = \lVert B_kx_k \rVert_2^2 + \lVert B_{k+1}^\top x_k \rVert_2^2
>   $
>
> > @ "Questions 2":
>
> As we replied above, MPSN does not have higher-order convolutions. Moreover, it has three times more parameters than SCCNN, which leads to more training data is required. As for the forex problem, it failed because it is not Hodge-aware, not able to capture the characteristic of the underlying edge flows. We have this in line 326 but will make this even more explicit. Hope this clarifies the question.
>
> > @ "Limitations..."
>
> We understand this may be a hard to follow paper because it provides several theoretical insights on a particular subfield. We hope the above points have clarified a few. We remain open to discuss more doubts that the reviewer finds difficulties.
>
> With this in place, we would like to clarify that the paper has no technical flaws (none of the reviewers pointed to such) or a poor evaluation. Therefore, we find a score of 2 based on writing style and a few grammar errors overly harsh. We remain open to discuss more doubts that the reviewer finds difficult.

---

> > ### Author Response · Authors · 2023-08-12
> >
> > A correction: The SCF is defined in Line 112. (Sorry for the confusion)

---

> > ### Comment · Reviewer_d7bL · 2023-08-20
> > **Thank you for the response**
> >
> > Thank you for the detailed response.
> >
> > I read through the response and again going through the paper, I am still of the opinion that the paper is a special case of MPSN or trivial adaptation of MPSN by removing few aggregation terms from the neighbourhood (please see discussion below Equation (11) in the MPSN paper). The related neural models like [15] with an orthogonal subspace decomposition assumption boils down to the models in [31], [16], [14], [19]. This makes the main contribution very limited or straightforward for the venue.
> >
> > The question about, why the discussed results or theory on expressiveness is important, remains. It is typical to characterize the expressiveness in terms of the simplicial Weisfeiler-Lehman test. Which could be something interesting.
> >
> > As far the experiments are concerned, it is again very limited. How does this model perform on clique-lifted graph tasks or large-scale datasets tasks. Many standard datasets (contact networks, clique-lifted graph datasets, collaboration network) do not exibit the orthogonal subspace decomposition, which makes MPSN more generic.
> >
> > Keeping these in mind, I will retain the previously given scores.

---

> > > ### Author Response · Authors · 2023-08-20
> > > **Reply part 1/3**
> > >
> > > Thanks for your reply. The reviewer made three points. Please see our response below.
> > >
> > > > @ #1: "the paper is a special case of MPSN or trivial adaptation of MPSN...The related [15] .. boils down to ... main contribution very limited or straightforward.."
> > >
> > > Equation (11) in MPSN [17] is a particular convolutional instance of MPSN by using the Hodge Laplacian shifting $L_kX_k$ as the message passing from direct simplicial neighbors and using $B_k^\top X_{k-1}$ and $B_{k+1}X_{k+1}$ as the message passing from faces and cofaces. This is essentially the same as [14].
> > >
> > > To help the reviewer see the difference, we made the following table listing the __topological differences__ between these methods. If it helps, please refer to Appendix Figure B.1 for a visual illustration about how these three principles act on simplicial signals.
> > >
> > > |Method|Scheme|Uncouple lower and upper simplicial adjacencies?|Inter-simplicial Coupling?|Multi-hop Interaction?|
> > > |-|-|-|-|-|
> > > |MPSN [17]|message-passing|yes|yes|no, only direct|
> > > |Eq. (11) of MPSN, or [14]|convolutional|no|yes|no, only direct|
> > > |Eq. (27) of MPSN|convolutional|yes|yes|no, only direct|
> > > |SNN [15]|convolutional|no|no|yes|
> > > |PSNN [16]|convolutional|yes|no|no, only direct|
> > > |SCNN [19]|convolutional|yes|no|yes|
> > > |This paper|convolutional|yes|yes|yes|
> > >
> > > On the one hand, by solely comparing these methods, indeed one generalizes or particularizes another. They do vary in terms of small aspects, either message-passing or convolutional, interacting with either direct or long-range neighbors, and so on.
> > >
> > > On the other hand, one can be intrigued by the flexible design choices induced by the simplicial topology and algebra. For example, _which scheme is better, convolution or message-passing? On what tasks? Why certain choices of convolutions give huge improvement? Why minor difference in the convolution methods can leads to disastrous failure on certain tasks?_
> > >
> > > _For these very two reasons, the main contribution [page 2] of this paper does __Not__ focus on proposing a new method SCCNN, but lies in theoretically advocating the choices of SCCNN (the analyses in Sections 3.2, 4 & 5)._ Particularly, for the purpose of learning from simplicial signals, the paper answered the following questions:
> > >
> > > >> 1. _why is Hodge-aware learning important?, compared to the non-Hodge-aware MPSN._
> > >
> > > The main difference lies in whether the method exploits the Hodge decomposition, i.e., whether it respects the three properties in Theorem 7. The Hodge decomposition is __not an assumption__ but a __universal__ decomposition that __all__ simplicial signals admit. Some data, e.g., forex, does show explicit characteristics by presenting only in one or two subspaces, particularly physical data. Some data, e.g., collaboration data, does not show such properties.
> > >
> > > In the task of learning simplicial signals, both perform learning a mapping $:V\to V$ with the simplicial signal space $V=\mathbb{R}^{n_k}$. Universally, $V$ admits the Hodge decomposition $V=G\oplus C\oplus H$ where $G,C,H$ are the gradient, curl and harmonic spaces respectively.
> > > - The Hodge-aware learner $T_1$ exhibits that
> > >     1. $G,C,H$ are invariant subspaces w.r.t. $T_1$. That is, $T_1 x \in G$ if $x\in G$. Likewise for if $x\in C$ and $x\in H$. Then, we can write $T_1(V) = T_1|_G(G) \oplus T_1|_C(C) \oplus T_1|_H(H)$ where $T_1|_G:G\to G$ is the restriction of $T_1$ on $G$, which is a mapping of a smaller space $G$. Likewise for the restrictions of $T_1|_C$ and $T_1|_H$, which are mappings between small spaces. This substantially shrinks the learning space, allowing for effective learning. (This corresponds to the convolutional choice.)
> > >     2. The learning in $G$ and $C$ are independent. That is, $T_1|_G$ and $T_1|_C$ have different learnable parameters, which do not depend on each other. (This corresponds to the choice to separate the lower and upper adjacencies in the Hodge Laplacians.)
> > >     3. The learnable mapping is expressive enough such that $T_1|_G$ can approximate any analytical functions from $G$ to $G$, and likewise for $T_1|_C$. (This corresponds to the expressive power in Theorem 6.)
> > > - MPSN $T_2$ using MLP to aggregate messages is non-Hodge-aware: $G,C,H$ are not invariant subspaces of $T_2$. Then, $T_2x\in V$ if $x\in G$, that is, $T_2$ would "mix" the subspaces and we do not have the expression as $T_1(V)$. This
> > >   - requires a large learning complexity, e.g., in simplex prediction, MPSN requires three times more parameters than SCCNN with comparable performance (Lines 348-349).
> > >   - does not preserve the subspaces. This is critical for signals presenting explicit Hodge characteristics. This is why MPSN gave unfair forex rates with large arbitrage, though small mean-squared-errors.

---

> > > > ### Author Response · Authors · 2023-08-20
> > > > **Reply part 2/3**
> > > >
> > > > - Without the multi-hop interaction, one can conclude MPSN is more general than SCCNN. However, this does not necessarily make MPSN a good learner for simplicial signals. Instead, SCCNN exploits _the inductive bias provided by the universal Hodge decomposition_, allowing for more effective learning.  Moreover, _the link (Eq.(4)) between the simplicial topology and Hodge spectra is another advantage of the SCCNN which is not the case for MPSN._ Informally, if one views MPSN as a black-box learning method on simplicial complexes, the Hodge-aware learner, built upon the Hodge decomposition, is much less of a black-box.
> > > > - __All in all, we view the MPSN and SCCNN as complementary/alternative solutions on simplicial complexes, each coming with its pros and cons. For example, MPSN has analyzable expressive power in terms of simplicial WL test, which is guaranteed for graph/complex classifications, while SCCNN has analyzable Hodge-awareness, which is guaranteed for learning simplicial signal tasks. More than putting them one against another, this paper advocates why SCCNN is a valid choice for learning simplicial signals.__
> > > >
> > > >
> > > > >> 2. _Why should one separate the two simplicial adjacencies but not combine them?_
> > > > - Separating the two adjacencies may appear as a trivial extension of not separating them, which leads to greater expressive power due to the increase of degree of freedom. In the message-passing framework, this is the explanation.
> > > > - However, in the convolutional flavor, this is necessitated by the fact that the two adjacencies encoded in the lower and upper Hodge Laplacians operate in two orthogonal subspaces, gradient and curl spaces, given by the Hodge decomposition. Spectrally, the learning functions in the gradient and curl frequencies are independent. In all three experiments, we see consistent improvements of doing so.
> > > > - The dangerous pitfall of not doing so: two simplicial signals located in two different subspaces but share the same frequency value cannot be learned separately. This explains __why SNN gave disastrous results in forex__ with large mean-squared-error and large arbitrage. It cannot separate the arbitrage as noise from the desired forex rates, as they share a common frequency value despite located in different subspaces.
> > > >
> > > > >> 3. _Why should one consider higher-order convolutions?_
> > > > - A direct advantage: higher-order convolutions allow for long-range interactions with multi-hop  simplicial neighbors, multi-hop faces and cofaces. This increases the expressive power.
> > > > - Although methods with direct interactions stacked multiple layers can increase the interaction range as well, the stability analysis showed us that to mitigate the mutual stability dependence between learning on different levels of simplices, one should consider shallow layers. To maintain the expressive power, higher-order convolution is a good principle in this sense. (Section 5)
> > > > - They are direct analogous of the convolution operator in the Euclidean domain (for time series or images) and on graphs [51 and refs thereof]. As such, they allow for (polynomial) filtering in the spectrum with a spectral interpretation adapted to the Hodge subspaces.
> > > >
> > > > >> 4. _Why should one account for the inter-simplicial couplings?_
> > > > - The Hodge theory states that the lower contribution contains information in the gradient space while the upper contribution contains information in the curl space. We can exploit them.
> > > > - It helps mitigate the simplicial oversmoothing. (Section 3.2)
> > > >
> > > > All above claims cannot be made without the theoretical tractable analyses in Sections 3.2, 4 & 5, which we claim as the main contributions and have not been done by existing works, not the architecture itself.
> > > >
> > > > > @ #2 "..Why the discussed results or theory on expressiveness is important?... It is typical to characterize the expressiveness in terms of the simplicial Weisfeiler-Lehman (WL) test..."
> > > >
> > > > The goal of studying expressive power is relative to the task.
> > > > For GNNs, one can use spectral graph theory, graph WL test, probabilistic graphical models, even the number of linear regions as done in [17, sec. 5]. One cannot say the number of linear regions analyzed in [17] is not important just because the WL test is widely adopted.
> > > >
> > > > If our task is to perform graph classification or simplicial complex classification, we shall perform expressive power analysis from the perspective of WL test. But that is not our goal here and restricting to an expressive analysis just because it is typical does not help much in research in general.
> > > >
> > > > As in our reply above, the expressive power in Theorem 6 is meant to show the Hodge-aware learner restricted in any subspace can express any mappings within this subspace. That is to show $T_1|_G$ can express any analytical functions between $G$s and $T_1|_C$ can express any analytical functions between $C$s. This is important for the goal of _learning simplicial signals_, _not_ for _classifying simplicial complexes or graphs_.

---

> > > > > ### Author Response · Authors · 2023-08-20
> > > > > **Reply part 3/3**
> > > > >
> > > > > > @ #3 "...experiments...very limited. How does the model perform on clique-lifted graph tasks or large-scale graph tasks?...Many standard datasets do not exhibit the orthogonal subspace decomposition..."
> > > > >
> > > > > We believe this is a subjective opinion by the reviewer. We included three experiments, which are real-world application driven. All involve with learning simplicial signals, which serves its purpose to corroborate the theory.
> > > > > - Learning forex rates involves learning edge flows, which was done also in [11] by optimization and requires both accuracy and arbitrage-free. Though we are the first applying simplicial signal learning to this task, it does not make the task limited.
> > > > > - Simplex prediction is an extension of link (1-simplex) prediction, which is one of the main graph tasks. We designed the experiments of 2-/3-simplex predictions by generalizing [52] and we chose the collaboration dataset preprocessed by [15].
> > > > > - Trajectory prediction was originally designed by [16], which was the first work applying learning simplicial signals to this task. This experiment does not appear to be limited in [16].
> > > > >
> > > > > About other dataset and tasks mentioned by the reviewer.
> > > > >
> > > > > - "on clique-lifted graph tasks": First, this is not the goal of the paper. We are learning simplicial signals residing on simplicial complexes which have simplices $n_k\ll\mathcal{O}(n^{k+1})$ in analogy to graphs having edges $\ll\mathcal{O}(n^2)$. This has been clearly stated in Introduction Lines 27-29, 44-47, in last Section Lines 393-403, as well as through the analyses. Second, we do not perform graph tasks by lifting graphs to clique complexes due to the complexity issues. The number of $k$-simplices has the worst exponential complexity $\mathcal{O}(n^{k+1})$ with $n$ the number of nodes. This has been discussed in [17, page 5] and [61, page 10].
> > > > > - "on large-scale dataset tasks": First, being practical, while large-scale datasets (e.g., obg datasets) are available for graph classification tasks, we are not aware of any large-scale simplicial signal learning tasks up-to-date. Second, adding such datasets (small or large scale) does not alter our messages here, as the analyses do not depend on the scale of dimensions. For example, in simplex prediction, there are 3.3k 2-simplices and 5k 3-simplices, where we observe consistent performance. Also, the scale of trajectory datasets in [16] does not affect the message of [16], nor our paper. Third, the scalability is much less of a concern for our method than for message-passing method due to the convolutional nature of the method. This is in analogy to the graph message passing and graph convolution networks. Note that SCCNN has computational complexity linear to the dimensions of simplices $n_k\ll\mathcal{O}(n^{k+1})$, and parameter complexity linear to the convolutional order.
> > > > > - "many standard datasets do not exhibit the orthogonal subspace decomposition, which makes MPSN more generic":
> > > > > 1. First of all, __all simplicial signals admit the Hodge decomposition__. We assume the reviewer meant that "many standard datasets do not present explicit characteristics of living in certain Hodge subspaces". We discussed this in Lines 399-403. Also, in our reply to #1, for simplex prediction in collaboration networks, such explicit properties are not present. However, this does not mean the Hodge-aware learning is not useful. Instead, it performs invariant learning in the three subspaces, substantially shrinking the learning space promoting the efficiency. Particularly, in simplx prediction, SCCNN achieves comparable performance to the costly MPSN, which has three times more parameters than SCCNN for 2-simplex prediction (Lines 348-349).
> > > > > 2. Moreover, many other datasets do present such properties. For example, The Hodge decomposition has been used to characterize traffic flows [11], brain networks [12], exchange market [24], statistical ranking [24], game theory [25], electrical networks [29] and so on.
> > > > > 3. The three experiments in this paper are less popular than the GNN datasets/tasks, but this does not make them limited. Instead, they are rather exploratory, and show the potentials of simplicial complexes instead of being limited to the application of graph classifications.
> > > > >
> > > > > We hope this clarifies the take of the paper, the significance of its contribution, and its importance in a multitude of highly-important tasks that go outside of the bubble of graph classification, standard datasets and WL tests.

---

### Official Review · Reviewer_KR9e · 2023-07-26

**Soundness:** 3 good
**Presentation:** 2 fair
**Contribution:** 3 good
**Rating:** 7
**Confidence:** 2

**Summary:**

This paper introduces a novel architecture designed to operate on simplicial data, drawing its foundation from the Hodge decomposition. This decomposition ensures that features associated with a simplicial complex of order $k$ can be represented by three distinct quantities: a curl-free quantity, a divergence-free quantity, and a harmonic quantity.
The authors develop SCCNN, a new architecture for simplicial data that abides by these decomposition principles. The authors demonstrate the relevance of these principles by examining the Dirichlet energy, serving as a measure of oversmoothing within simplicial networks. The authors show that this new architecture reduces oversmoothness.
Theoretical guarantees for the stability of the proposed method when subjected to small perturbations are proven. The proposed approach is benchmarked against two exploratory tasks: a forex test and simplex prediction. The findings indicate superior performance of the SCCNN over previous architectures in these tasks.

**Strengths:**

- This paper delivers a substantial theoretical advancement to the field of neural networks operating on simplicial complexes. It introduces compelling theorems within a well-structured framework. Theorem 6. on the expressiveness of the SCCNN model is a result of potentially good interest for this subfield.
-  The authors have thoroughly compared their approach to existing methods, including standard graph neural networks (GNNs) and other simplicial neural networks. The results on toy examples are compelling.
- The paper does a good job of introducing the key concepts in a clear and precise way. I appreciated the background on Hodge decomposition.
- Due to the simplicity of the overall principle behind SCCNN,  it has the potential for broad application within the field and formalizes new fundamental design principles for future architectures.

**Weaknesses:**

- The paper will benefit from enhanced clarity. Currently, it contains abundant results, which, while potentially insightful, obscure the core message of the research. Streamlining these results and focusing on the most salient points would aid in transmitting the core of the research, which is the SCCNN architecture, more effectively. The discussion around stability, while interesting, feels out of place. There are no clear motivations for studying it so thoroughly in the main text. On the other hand, the critical Theorems, such as Theorems 6. and 7., will benefit from a lengthier exposition. In particular, giving intuition behind the proofs of Theorem 6 would be appreciated. While I appreciate it is a theoretical paper, the results go in every possible direction with no clear target. In such a short paper, one should focus on a few key ideas and move as much as possible to appendices.

- The forex example is somewhat tailored to align with the proposed method. While it proves the paper's point and should be kept, it feels too synthetic. I am happy to be contradicted by the authors on that.

**Questions:**

- Could the authors provide more insight on this sentence in the limitations: SCNN "cannot learn differently from features at the
frequencies of the same type and the same value". I find the sentence very confusing. For example GNNs, can learn different node features over different channels, while they are all harmonic features.
- I would like to understand why stability deserves such lengthy exposition in this paper. Currently, it feels out of place. What makes the stability of higher importance for SCCNN than for other architectures? It would deserve more experiments to show why it is essential in networks on simplicial complexes.

**Limitations:**

Limitations are discussed. Some points need to be clarified. See my question above.

---

> ### Author Rebuttal · Authors · 2023-08-04
>
> We really appreciate the concise summary of the paper from the reviewer!
>
> > @ Weakness #1: about why we included stability analysis and further exposition of Thms 6 and 7.
>
> We appreciate your thorough suggestions. Following your suggestion, we will focus the results on the main points (comparisons, corroboration) and move to the appendix the "ablation study" and the "convolution orders on stability". We also made the text about the stability more concise and with the overall gained space, we elaborated on the consequences and insights from Thms 6 and 7.
>
> The motivation to study the stability of neural network (NN) solutions to domain perturbations is rooted in the early works from Bruna and Mallat [43], which measures the ability of a NN to (i) understand the role of each component in robust to small adversaries; (ii) discuss their ability to transfer to a similar slightly-changed domain, and if so, quantify when this is possible and when not. This has also been widely studied for GNNs [46-49] but there are no results in the SC domain. Please also refer to our reply to _Question #2_.
>
> In terms of detailed actions,
> - we have included an illustration figure in the PDF in _Global Response_ to convey the Hodge-aware idea of Theorem 7 and included more discussions as in our _Global Response_ and _Response to reviewer XgBj_.
> - we included a sketch proof leading to the results in Theorem 6 before the spectral interpretation in Lines 201-204.
>
> > @ Weakness #2: about forex, an example is too synthetic.
> - We appreciate your concern about this. The forex example may appear to be too synthetic in the sense that it has a very identifiable characteristic, i.e., being curl-free. This is however a practical need corresponding to the arbitrage-free exchange market. Such characteristics arise often also in physical problems, e.g., water flows being divergence-free, electric current being divergence-free while electric voltage being curl-free, or electrical and magnetic fields. We reasonably foresee the applications of learning on simplicial complexes to such physical problems, as well as in other applications like statistical ranking [24] and brain networks [12].
> - We took real data for this experiment and added different kinds of noises and masked 50% of the values to create several challenging scenarios. When applying SCCNN to this task, _we did not add any assumptions on the curl (we were blind to the type of data and let the network figure it out)_. This is in comparison to the baseline method where a regularization is added to penalize the curl. We do observe impressive improvements in both MSE and arbitrage senses. This shows SCCNN can capture the characteristics of simplicial signals.
>
> > @ Question #1: about our discussion on the limitation.
>
> Thanks for raising this question about the limitation of the method. In the case of multiple features, the reviewer is correct and the SCCNN can indeed learn differently from these features even at the same type frequency of the same value. The claim here is w.r.t. a single specific feature. We made this aspect explicit in the manuscript now.
>
> > @ Question #2: about the importance of stability analysis
>
> When it comes to the stability analysis for SCCNN, there are both practical and theoretical motivations.
>
> 1. Practically, when implementing SCCNNs to say water/power/brain networks to process their flows or higher-order signals, we know the weights of the underlying topology up to uncertainty [Lines 249-253]. Therefore, the model will be trained on a different topology from the one that could be implemented. The stability results allow us to measure the robustness of the outputs. This practical motivation is similar to that for GNNs [46-49]. Another common motivation is that the architectures could often be under adversaries in their topological weights and identifying the handle against them is paramount.
>
> 2. From a theoretical perspective, studying stability of SCCNN is more challenging than GNN because the topology involves with the inter- and intra-simplicial couplings. This highlights how these couplings play a role in propagating the perturbations across the layers. As shown in Theorem 12, the stability of SCCNN concerns the outputs of all levels of the simplices and there exists a mutual negative influence between them. As we have experimentally shown in Figure 1(c), the perturbation on the graph Laplacian can influence the outputs on edges and triangles depending on the number of layers.
>
> 3. As of the results in Theorem 12, an implication is that one shall consider shallow layers for SCCNN to avoid the negative dependency between outputs on different simplicial levels. On the other hand, to maintain expressive power, higher-order convolution is a good choice, as demonstrated in Figure 4. This stability analysis also motivates our design of SCCNN.
>
> 4. The lengthier exposition (a bit more than a page) is needed here to formalize the concepts, make the assumptions explicit in the main body, and elaborate on the consequences of the main theoretical results. We have these results as adding to the ongoing stability research for neural networks (CNNs and GNNs mainly), which is the first-ever stability result for a model operating on a simplicial complex, especially under the case that SCCNN subsumes several other works. We find this rather a pioneering finding in this subfield.
>
> 5. Experimentally, besides the above main messages shown in Figures 1(c), 3, and 4, we also provided a detailed analysis in Appendices F.3.7 and F.4.4 where the stability in terms of the number of layers, the order convolutions and the mutual dependence are provided.
>
> We hope this more detailed explanation can help the reviewer to see the importance of stability analysis for SCCNNs. We really appreciate your suggestions on improving the paper.

---

> > ### Comment · Reviewer_KR9e · 2023-08-12
> >
> > Thank you for your answers.
> >
> > I appreciate your detailed response and proposed changes. I keep my original score. I encourage the authors to improve the readability of their paper and streamline their results.

---

### Official Review · Reviewer_xwZ6 · 2023-07-27

**Soundness:** 3 good
**Presentation:** 3 good
**Contribution:** 3 good
**Rating:** 5
**Confidence:** 2

**Summary:**

The authors propose a general convolutional architecture with principles of uncoupling the lower and upper simplicial adjacencies, which accounting for the inter-simplicial couplings, and performing higher-order convolutions for learning of higher-order structure and simplicial signal.

**Strengths:**

- The authors show that the proposed SCCNN structure demonstrates awareness of the Hodge decomposition and performs efficient learning on simplicial data
- Effect on mitigating simplicial complex oversmoothing is explained with Dirichlet energy minimization
- Comprehensive experimental results
- Stability against robustness is also studied in this work

**Weaknesses:**

See questions

**Questions:**

- How is the orientation of simplex determined in the experiments?
- In the definition of simplicial data / k-signal, is it defined for one simplex or a k-chains? What are their dimensions? Is it similar to a feature vector or it is changing over time? It would be helpful if the authors show an example of it the k signal and related terms, as one of the main claim is that "advantage of using SCs is not only about them being able to model higher-order network structure, but also support simplicial data"
- There are some typos in the paper, for example, line 395 "approcahed".
- For the simplex prediction problem, performance and baseline comparison with higher-order simplices (order=4,5,6...) could be also provided.

---

> ### Author Rebuttal · Authors · 2023-08-04
>
> We appreciate reviewer's comments on the strengths of our paper. Here we address your four detailed questions.
>
> > @ #1 about the orientation
>
> The orientation of a simplex is set as the lexicographical ordering of its vertices (line 79), as commonly done in simplicial literature [15, 16, 17, 27, 41]. For example, we defined the reference orientation of an edge $e=(i,j)$ as $i\to j$, and that of a triangle $(i,j,k)$ as $i\to j\to k\to i$.
> Specifically, in the experiments, we first label the vertices, then give the edge and triangle orientations based on the labels of their vertices.
> For example, in forex problem where vertices are currencies, for an edge $\{i,j\}$, we define the reference orientation as $[i,j]$ with the signal $x_{[i,j]}=\log(r^{i/j})$, the logarithm of the exchange rate from $i$ to $j$. If $x_{[i,j]}>0$, then $r^{i/j}>1$; otherwise $r^{i/j}\leq 1$.
> For simplex prediction, we followed the construction as in [15], while for trajectory prediction, we followed the construction in [16, 17].
>
> Note that the choice of the reference orientation is for computational purpose and inconsequential to the results if the method is orientation equivariant, because the orientation of a simplex is an equivalence class, saying that the orientations $[i,j,k]$ and $[j,k,i]$ are equivalent (aligned), while $[i,j,k]$ and $[j,i,k]$ are anti-aligned.
> We refer to [9] for a detailed discussion and [41, sec 2.3] for a slim discussion.
>
> (We also refer to Appendix B.3 (Lines 684-699) for our study of the orientation symmetry of simplicial signal space using the group theory, if the reviewer is interested.)
>
> > @ #2: about the definition of simplicial data
>
> A $k$-simplicial signal $x_k$ is defined via an alternating function mapping each $k$-simplex to a real-valued scalar. Thus, a $k$-simplicial signal $x_k$ has dimension $n_k$, the number of $k$-simplices (line 87). So it is defined for a $k$-chain. It is also possible to be viewed as a feature vector of the $k$-simplices. In the case of $d$ multiple features, we then have $X_k\in\mathbb{R}^{n_k\times d}$. We refer to Figure A.1 in Appendix A where we illustrated an edge flow together with other notions, including divergence, curl and the Hodge decomposition. We followed the common definition when processing and learning from simplicial signals; see e.g., [9, 15, 16, 17, 18, 27]. Please let us know if it is unclear.
>
> > @ #3: about the typo
>
> Thanks, fixed. We also did thorough proofreading to fix a few other typos.
>
> > @ #4: About 4-, 5-.. simplex prediction
>
> We appreciate the suggestion for additional experiments on simplex prediction for higher-order simplices. However, we believe that the current results on 2- and 3-simplices are sufficient to validate the model and corroborate the theoretical claims. Also, due to the page limit, we opted to provide results on other diverse experiments to show the potential applications of the method. Moreover, we ran a comprehensive ablation study, included in Appendix F.3. In terms of experiments on higher-order simplices (e.g., 4, 5,...), we believe such practical needs might be found in the field of topological methods for brain activity research [Ref 1], which is a potential future work on the application of learning methods on simplicial complexes.
>
> [Ref 1] Reimann, Michael W., et al. "Cliques of neurons bound into cavities provide a missing link between structure and function." Frontiers in computational neuroscience 11 (2017): 48.
>
> > @ our summary of this rebuttal
>
> As a final remark, we hope our answers adequately address the four questions posted by the reviewers. We notice that the first two questions concern detailed aspects regarding simplicial complexes and simplicial signals. While we made an effort to provide as many precise details on the preliminaries, it is often rather hard to cover every detail within a conference paper. Instead, we concentrated on the primary contributions of this paper, as listed also by the reviewer, while avoiding excessive mathematical jargon.

---

> > ### Comment · Reviewer_xwZ6 · 2023-08-18
> >
> > I thank the authors for their response.

---

### Author Rebuttal · Authors · 2023-08-04

In this global response, we first express our gratitude to the reviewers for providing us with their insightful and critical feedback.
The positive feedback from the reviewers acknowledges the strengths of our paper in the following areas:
- Studying an important problem: Our paper delves into the well-motivated learning from simplicial signals (shzQ), drawing its foundation from the Hodge theorem (XgBj, KR9e).
- Great overview of the simplicial complex and Hodge decomposition: Our paper does a good job of introducing the key concepts in a clear and precise manner (XgBj, KR9e, shzQ).
- Substantial theoretical contributions: Our work brings significant advancements to the field (KR9e), including the relevance of the principles behind SCCNN on reducing the simplicial oversmoothness via Dirichlet energy explanation (xwZ6, XgBj, KR9e), the Hodge-awareness of SCCNN (xwZ6, XgBj, KR9e), and the theoretical analysis of the stability against domain perturbations (xwZ6, XgBj, KR9e).
- The theoretical claims are backed by exploratory experiments (KR9e) on synthetic and real SC datasets (XgBj) with comprehensive comparisons to existing methods (KR9e, xwZ6).
- Due to the simplicity of the overall principle behind SCCNN, it has the potential for broad application within the field and formalizes new fundamental design principles for future architectures. (KR9e)

We are encouraged by these comments.

Secondly, while we hope our individual replies address reviewers' specific questions, we here draw our attention to the feedback by _Reviewers KR9e and XgBj_ that a lengthier exposition on the Hodge-awareness (Thm 7) would be helpful.

We gave the thumbnail properties of Hodge-aware learning in Thm 7 where we omitted the rigorous formulation of each Hodge subspace being invariant under the learning operators.
While these properties follow from eq (4) (a spectral description of Hodge-aware learning) and the expressive power in Thm 6, we followed two reviewers' suggestions. To better convey the idea, we have included an illustrative figure in the _attached PDF_ and enriched the discussions in L211-224 with the following.

The definition of Hodge-aware learning is composed of three _theoretically analyzable_ properties:
1. The three Hodge subspaces are __invariant__ under the learnable $H_k$ of SCCNN: Consider learning from a simplicial signal $x_k$.
- Its gradient component can be only learned via the restriction of $H_k$ in $\text{im}(B_k^\top)$, denoted $H_{k}|_{\text{im}(B_k^\top)}:\text{im}(B_k^\top)\to\text{im}(B_k^\top)$, because the gradient space is invariant w.r.t. $H_k$.
- Its curl component can be only learned via the restriction of $H_k$ in $\text{im}(B_{k+1})$, denoted $H_{k}|\_{\text{im}(B_{k+1})}:\text{im}(B_{k+1})\to\text{im}(B_{k+1})$, because the curl space is invariant w.r.t. $H_k$; and likewise for its harmonic component learned via $H_k|_{\text{ker}(L_k)}$. Also, the lower (and upper) contributions having only gradient (and curl) component is learned via an operator to which the gradient (and curl) space is invariant.
- In short, given the Hodge decomposition of the simplicial signal space $\mathbb{R}^{n_k}=\text{im}(B_k^\top) \oplus\text{im}(B_{k+1}) \oplus \text{ker}(L_k)$, the learning operator $H_k:\mathbb{R}^{n_k}\to\mathbb{R}^{n_k}$ applied to the signal space can be written as
$H_k(\mathbb{R}^{n_k}) = H_{k}|\_{\text{im}(B_k^\top)}(\text{im}(B_k^\top)) \oplus H_{k}|\_{\text{ker}(L_k)}(\text{ker}(L_k)) \oplus H_{k}|\_{\text{im}(B_{k+1})}(\text{im}(B_{k+1})).$
2. The learnable operators are __independent__ in the nontrivial gradient and curl spaces: The learning of $H_k$ in the gradient and curl spaces is controlled by the learnable weights $w_{k,\rm{d},t}$ and $w_{k,\rm{u},t}$, respectively (likewise for the learning of $H_{k,\rm{d}}$ and $H_{k,\rm{u}}$). Spectrally, it induces two independent spectral responses for gradient and curl frequencies (eq (4)).
3. The learning in the gradient and curl spaces is __expressive__. Spatially, a SCCNN layer is as expressive as an MLP with certain transformation operators (Thm 6). Spectrally, the learning curves are as expressive as analytical functions in both gradient and curl subspaces, as discussed in L 200-205 and shown in Figure 1(a).

However, what is the point of these properties?
1. The invariance property substantially shrinks the learning space, simply because it allows us to decompose the learning operator into operators defined on three smaller Hodge subspaces. Consider a counterexample where gradient space is not invariant w.r.t. the learning operator $M$. Then, $M$ would map the gradient space to any other subspace, i.e., would ``mix $\text{im}(B_k^\top)$ with $\text{im}(B_{k+1})$''. This would result in a much larger learning space. MPSN [17] is such an example, which uses MLP to aggregate and update information on simplicial complexes.
2. The independent learning in two subspaces intuitively increases the expressive power due to the use of independent learnable weights in the gradient and curl spaces. Moreover, this is however often critical for learning simplicial signal tasks: When a gradient frequency equals a curl frequency, if a non-independent learning is applied, then the features at this frequency but in two spaces are always learned in the same manner. SNN [15] and Bunch [14] are two counterexamples.
3. While respecting the first property can reduce the learning space, it can also limit the expressive power of a learning operator. We need to have expressive learning operators such that they can approximate any learning curves in the corresponding frequencies of each subspace (Thm 6).

Having these properties within a learning framework is tantamount to achieving independent and expressive learning within each Hodge subspace while maintaining their invariance, as illustrated in the PDF. This facilitates both effective and rational learning from simplicial signals (L 211-233) and we call it Hodge-aware learning.

---

### Decision · Program_Chairs · 2023-09-21

**Decision:**

Reject

**Comment:**

This paper proposes a simplicial complex convolutional neural network (SCCNN), analyzes its theoretical properties (over-smoothing, Hodge-awareness, robustness), and validates the empirical effectiveness.

This paper was very borderline for decision despite the low review scores (7/5/4/4/2).
- Reviewer d7bL expressed concerns about the low quality of writing and unclear contributions. Upon reading the paper and the rebuttal myself, I find the contributions to be clear and I believe the quality of the writing can be improved simply by incorporating the reviewer's comments.
- Reviewer shzQ and XgBj were mainly concerned about the limited application and evaluation of the proposed framework and pointed out possible comparisons on graphs. I resonate with the authors that SCCNN is about simplicial complexes and does not require evaluation on graphs. However, I agree with the reviewers that current experiments are insufficient to claim the usefulness of the SCCNN framework in practice. The authors considered two real-world datasets (ocean drift and forex), but as reviewer KR9e pointed out, I think the forex dataset is a little bit synthetic. The author's rebuttal was not convincing enough to fully resolve this issue.

Overall, I think this paper has good potential since it yields nice theoretical results about neural networks on simplicial complexes. However, the reviewers seem to be lukewarm about this work since its practical benefits are unclear. Also, the presentation could be improved for better readability, e.g., reviewers raised several questions about the core concepts in their initial review.

These are non-trivial issues that can not be easily alleviated, hence I recommend rejection for this paper. I believe this paper will become a very strong submission to the next conference after incorporating the comments.